# The Aarhus Chamber Campaign on Highly Oxygenated Organic Molecules and Aerosols (ACCHA): Particle Formation, Organic acids, and Dimer Esters from Alpha-Pinene Ozonolysis at Different Temperatures

K. Kristensen[1], L. N. Jensen[2], L. L. J. Quéléver[3], S. Christiansen[2], B. Rosati[2,4], J. Elm[2], R. Teiwes[4], H. B. Pedersen[4], M. Glasius[2], M. Ehn[3], M. Bilde[2]

[1]Department of Engineering, Aarhus University, 8000 Aarhus C, Denmark
[2]Department of Chemistry and iClimate, Aarhus University, 8000 Aarhus C, Denmark
[3]Institute for Atmospheric and Earth System Research – INAR / Physics, P.O. Box 64, FI-00014, University of Helsinki, Finland
[4]Department of Physics and Astronomy, Aarhus University, 8000 Aarhus C, Denmark

*Correspondence to*: Kasper Kristensen (kasper.kristensen@eng.au.dk) and Merete Bilde (bilde@chem.au.dk)

## Abstract

Little is known about the effects of sub-zero temperatures on the formation of secondary organic aerosol (SOA) from α-pinene. In the current work, ozone-initiated oxidation of α-pinene at initial concentrations of 10 ppb and 50 ppb, respectively, is performed at temperatures of 20 °C, 0 °C and -15 °C in the Aarhus University Research on Aerosol (AURA) smog chamber during the Aarhus Chamber Campaign on Highly oxidized multifunctional organic molecules and Aerosol (ACCHA). Herein, we show how temperature influences the formation and chemical composition of α-pinene-derived SOA with a specific focus on the formation of organic acids and dimer esters. With respect to particle formation, the results show significant increase in particle formation rates, particle number concentrations and particle mass concentrations at low temperatures. In particular, the number concentrations of sub-10 nm particles were significantly increased at the lower 0 °C and -15 °C temperatures. Temperature also affects the chemical composition of formed SOA. Here, detailed off-line chemical analyses show that organic acids contribute from 15 % to 30 % by mass, with highest contributions observed at the lowest temperatures, indicative of enhanced condensation of these semi-volatile species. In comparison, a total of 30 identified dimer esters were seen to contribute between 4 – 11 % to the total SOA mass. No significant differences in the chemical composition (i.e. organic acids and dimer esters) of the α-pinene-derived SOA particles are observed between experiments performed at 10 and 50 ppb initial α-pinene concentrations, thus suggesting a higher influence of reaction temperature compared to that of α-pinene loading on the SOA chemical composition. Interestingly, the effect of temperature on the formation of dimer esters differs between the individual species. The formation of less oxidized dimer esters (with oxygen-to-carbon ratio (O:C) < 0.4) is shown to increase at low temperatures while the formation of the more oxidized species (O:C > 0.4) is suppressed, consequently resulting in temperature-modulated composition of the α-pinene derived SOA. Temperature ramping experiments exposing α-pinene-derived SOA to changing temperatures (heating and cooling) reveal that the chemical composition of the SOA with respect to dimer esters is governed almost solely by the temperature at which oxidization started and insusceptible to subsequent changes in temperature. Similarly, the resulting SOA mass concentrations were found to be more influenced by the initial α-pinene oxidation temperatures, thus suggesting that the formation conditions to a large extent govern the type of SOA formed, rather than the conditions to which the SOA is later exposed.

For the first time, we discuss the relation between the identified dimer ester and the highly oxygenated organic molecules (HOMs) measured by Chemical Ionization Atmospheric Pressure interface Time-Of-Flight mass spectrometer (CI-APi-TOF) during the ACCHA experiments. We propose that, although very different in chemical structures and O:C ratios, many dimer esters and HOMs may be linked through similar $RO_2$ reaction pathways,, and that dimer esters and HOMs merely represent two different fates of the $RO_2$ radicals.

## 1. Introduction

The oxidation of volatile organic compounds (VOC) constitutes an important source of secondary organic aerosol (SOA) in the atmosphere. Due to its atmospheric reactivity and its high estimated global emission rate of ~30 Tg/yr (Sindelarova et al., 2014), the atmospheric oxidation of the biogenic VOC α-pinene has been widely studied. The boreal forests are considered abundant sources of α-pinene with varying but sizeable emissions occurring all year around (Hakola et al., 2003;Hakola et al., 2009;Noe et al., 2012). In addition, Lee et al. (2020) and Zhang et al. (2018) show that monoterpene originated SOA are the largest sources of particulate matter in the southeastern US. For this reason, the oxidation of α-pinene and subsequent formation of SOA is expected to occur at conditions with a wide range of VOC concentrations and atmospheric temperatures.

A number of smog chamber studies have investigated the effect of α-pinene concentration and reaction temperature on the formation of SOA from α-pinene oxidation (Svendby et al., 2008;Pathak et al., 2007;Tillmann et al., 2010;Warren et al., 2009;Kourtchev et al., 2016;Kristensen et al., 2017;Zhao et al., 2019a). Most of published studies, however, focus on SOA mass yield (defined as mass of SOA formed per mass of reacted VOC) and reports higher yields with increasing VOC concentrations and lower temperatures. The effect of subzero temperatures, on gas phase oxidation products, nucleation, particle growth and particle chemical composition, remains a largely unexplored area (Huang et al., 2018;Stolzenburg et al., 2018).

Once emitted to the atmosphere, α-pinene is oxidized through reactions with atmospheric oxidants such as ozone ($O_3$), hydroxyl ($OH^.$) and nitrate ($NO_3^.$) radicals. These reactions have been shown to result in numerous oxidation products with various chemical functionalities, including alcohols, aldehydes, ketones, carboxylic acids, peroxides and peroxy-acids (Nizkorodov et al., 2011;Noziere et al., 2015). As a result of their lower volatilities, many α-pinene-derived oxidation products (e.g. carboxylic acids) partition to already existing particles in the atmosphere, contributing to particle growth and increasing SOA mass. The extent and timescale to which an organic compound undergoes partitioning is related to its saturation vapor pressure, the available particle mass (Kroll and Seinfeld, 2008) as well as particle size and particle phase (Shiraiwa and Seinfeld, 2012;Li and Shiraiwa, 2019;Zaveri et al., 2014;Zaveri et al., 2020). Due to the temperature dependence of the saturation vapor pressures of organic oxidation products, higher SOA mass yields from increased condensation of organics have been observed at lower temperatures (Pathak et al., 2007;Saathoff et al., 2009;Warren et al., 2009;Svendby et al., 2008). The saturation vapor pressure of a given organic species is closely related to its chemical structure. Depending on their specific molecular structure and the amount of strong hydrogen bond donor-acceptor groups, the products of α-pinene oxidation may undergo condensation or, if sufficiently low volatile, nucleation, with the latter resulting in new particle formation (Kulmala et al., 2013;Riccobono et al., 2014).

In addition to temperature-dependent condensation of oxidation products, temperature-modulated gas and multiphase chemistry has been suggested to influence SOA yields from VOC oxidation. Von Hessberg et al. (2009) observed oscillatory positive temperature dependence under dry conditions and suggested that SOA yields from β-pinene oxidation is governed to a higher degree by the temperature and humidity dependence of the involved chemical reactions than by vapor pressure of the formed oxidation products at different temperatures. Furthermore, Pathak et al. (2007a) observed that α-pinene SOA yields showed a weak dependence on temperature in the 15 °C to 40 °C range, implying that the negative temperature dependence of the partitioning is counteracted by a positive dependence of the chemical reaction mechanism. Multiple studies report on new

particle formation arising from the oxidation of α-pinene and suggest the formation of so-called extremely low volatile organic compounds (ELVOC, Donahue et al. (2012b)) capable of gas-to-particle conversion (Bonn and Moorgat, 2002;Lee and Kamens, 2005;Gao et al., 2010;Tolocka et al., 2006;Claeys et al., 2009;Ehn et al., 2014;Metzger et al., 2010;Kirkby et al., 2016;Bianchi et al., 2019). Although the chemical composition of α-pinene SOA has been extensively studied, uncertainty still remains regarding the chemical structure and functional groups of the compounds responsible for nucleation. Recent quantum chemical results indicate that multi-carboxylic acids with three or more acid moieties are some of the most likely compounds to participate in new particle formation (Elm et al., 2017; Elm et al., 2019). In relation, Lawler et al. (2018) observed enhanced content of alkanoic acids in newly formed 20 –70 nm particles in the Finnish boreal forest.

Recently, highly oxygenated organic molecules (HOMs) have been identified in α-pinene oxidation studies (Bianchi et al., 2019;Ehn et al., 2014;Jokinen et al., 2014;Rissanen et al., 2015). Formed via a gas-phase autoxidation reaction (Crounse et al., 2013) involving intramolecular hydrogen abstraction by peroxy-radicals, HOMs represent a class of organic compounds which can promptly reach high degrees of oxygenation (Ehn et al., 2017). Oxygen-to-carbon (O:C) ratios exceeding unity have been reported for monomer compounds (Mutzel et al., 2015;Ehn et al., 2014). The high O:C ratios are attributed to multiple hydroperoxide functionalities and the compounds have been perceived as likely candidates for nucleation and initial particle growth owing to their low volatilities (Tröstl et al., 2016;Kirkby et al., 2016). However, computational studies have shown that HOMs originating from α-pinene autoxidation can have surprisingly high vapor pressures considering their oxidation state, and it is mainly the dimeric compounds that are likely to be classified as ELVOC (Kurtén et al., 2016;Peräkylä et al., 2020).

High molecular weight (MW > 300 Da) dimer esters formed from oxidation of α-pinene have been identified in laboratory-generated and ambient air SOA (Hoffmann et al., 1998;Tolocka et al., 2004;Gao et al., 2004;Yasmeen et al., 2010;Witkowski and Gierczak, 2014;Kourtchev et al., 2015;Zhang et al., 2015;Kristensen et al., 2013;Kristensen et al., 2014;Kristensen et al., 2016;Kristensen et al., 2017;Mohr et al., 2017). Several mechanisms for the formation of dimer esters have been proposed, including gas-phase mechanisms such as clustering of carboxylic acids (Hoffmann et al., 1998;Claeys et al., 2009a;DePalma et al., 2013) and reactions involving reactive intermediate radicals such as stabilized Criegee intermediates (sCI) and $RO_2$ species (Kristensen et al., 2016;Zhang et al., 2015;Berndt et al., 2016;Witkowski and Gierczak, 2014;Zhao et al., 2015;Kahnt et al., 2018).

The high abundance of dimer esters detected in freshly formed α-pinene derived SOA particles (Kristensen et al., 2016) suggests that these compounds may be important for the initial formation and growth of atmospheric particles; a hypothesis supported by saturation vapor pressure estimates reported in Kristensen et al. (2017) classifying dimer esters as ELVOC with vapor pressures suitable for gas-to-particle phase conversion at room temperatures. Furthermore, recent studies show that dimer esters and other oligomeric compounds in atmospheric aerosol are strongly correlated with cloud condensation nuclei activity thus suggesting a significant impact on climate (Kourtchev et al., 2016).

In contrast to dimer esters, few studies have identified particle phase HOMs and thus their fate upon gas-particle partitioning remains elusive. Recently, Zhang et al. (2017) identified highly oxidized monomers ($C_{8-10}H_{12-18}O_{4-9}$) and dimers ($C_{16-20}H_{24-36}O_{8-14}$) in α-pinene derived SOA particles using $Na^+$ attachment during electrospray ionization. The dimers identified by Zhang et al. (2017) show a somewhat different chemical formula and degree of oxidation than the gas-phase HOM dimers previously identified by Ehn et al. (2014), where the observed dimer composition was $C_{19-20}H_{28-32}O_{11-18}$. This can be attributed to decomposition of the hydroperoxides functionality of HOMs from processes such as photolysis, thermolysis and solvation, yielding alkoxy radicals, esters, and other moieties. The chemical formulas of the dimeric compounds in Zhang et al. (2017) show good resemblance to the dimer esters published in Kristensen et al. (2016), characterized as $C_{15-19}H_{24-30}O_{5-10}$. This raises the important question, how different types of identified dimers, including HOMs and dimer esters, are related via gas-particle partitioning, chemical reactions or other processes.

The Aarhus Chamber Campaign on HOMs and Aerosols (ACCHA) presented in this work was designed to elucidate the formation of HOMs and dimer esters from the dark ozonolysis of α-pinene at different temperatures. The effect of temperature and α-pinene concentration on SOA formation and composition on the formation of HOMs and dimer esters is investigated through a series of chamber experiments at temperatures prevailing at the latitudes of the boreal forests (Hakola et al., 2009) i.e. from 20 °C to -15 °C.

The results of ACCHA are presented in multiple publications. The elemental composition of the bulk α-pinene SOA by High Resolution Time-of-Flight Aerosol Mass Spectrometer (HR-ToF-AMS) is reported in the companion paper by Jensen et al (2019). A study presenting factor analysis of PTR-ToF-MS data includes a case study from the ACCHA campaign (Rosati et al., 2019). The formation of gas-phase HOMs as measured by nitric acid-based Chemical Ionization Atmospheric Pressure interface Time Of Flight (CI-APi-TOF) mass spectrometer is presented in the work by Quéléver et al. (2019).

In the current work, we present the effect of temperature and α-pinene concentrations on the formation and molecular composition of SOA particles formed from ozone-initiated oxidation of α-pinene. Specifically, we investigate the contributions of organic acids and dimer esters to α-pinene SOA formed at temperatures of 20 °C, 0 °C, and -15 °C and examine the changes in molecular composition arising from heating and cooling of SOA particles.

## 2. Method

### 2.1 Chamber experiments

Dark ozonolysis of α-pinene was conducted in the Aarhus University Research on Aerosol (AURA) smog chamber. The chamber is described in detail in Kristensen et al. (2017). In short, the AURA chamber consists of a 5 m$^3$ Teflon bag situated in a 27 m$^3$ temperature controlled cold room. The chamber temperature can be varied from -16 °C to +26 °C. The temperature and relative humidity (RH) are monitored in the center of the Teflon bag by a HC02 probe coupled to a HygroFlex HF320 transmitter (Rotronic AG, Switzerland). Additional instrumentations are situated in air-conditioned (at constant 20°C) laboratory surrounding the cold room. In the current study, the chamber was operated using dry purified air (active carbon, Hepa) at atmospheric pressure. Ozone ($O_3$, ~100 ppb) was added to the chamber using an ozone generator (Model 610, Jelight Company, Inc.) and the concentration of $O_3$ and oxides of nitrogen (NO and $NO_2$) were monitored by UV photometric (O342 Module, Environment S.A.) and chemiluminescent monitors (AC32M, Environment S.A.), respectively. A known amount of VOC was added to a 10 mL glass manifold, evaporated and transferred to the chamber through a stainless-steel inlet (Diameter = 10 mm, Length = 100 cm) using heated (70 °C) $N_2$-flow. The concentration of the added VOC was monitored using a Gas Chromatograph with Flame Ionization Detector (Agilent 7820A GC-FID, with a time resolution of 6 min) and a Proton Transfer Reaction Time-of-Flight Mass Spectrometer (PTR-ToF- 783 MS 8000; IONICON, Innsbruck, Austria) sampling through stainless steel outlets located opposite of the VOC inlet (distance between inlet and outlet ~ 1.6 m). Particle size distributions were measured using a scanning mobility particle sizer (SMPS) system including a Kr-85 neutralizer (TSI 3077A) and an electrostatic classifier (TSI 3082) coupled with a water-based nano condensation particle counter (CPC, TSI 3788). The SMPS system was optimized for measurements of particles in the range of 10 – 400 nm with a sampling time of 80 s (60 s upscan, 20 s downscan, aerosol flow rate = 1.6 L min$^{-1}$, sheath flow rate = 5 L min$^{-1}$). In addition, the total particle number concentration of particles with diameter ($D_p$) larger than ~1.4 nm was measured by a Particle Size Magnifier (PSM A10, Airmodus, (Vanhanen et al., 2011), sample flow = 2.5 L min$^{-1}$, time resolution = 1 s) operated in fixed saturator flow mode. SMPS and PSM measurements were performed as close as possible to the cold room trough insulated tubing extending ~ 40 cm and ~ 10 cm from the cold room, respectively, thus minimizing residence time and potential influences caused by temperature variations.

The chemical composition of gas-phase HOMs were measured using a nitric acid-based Chemical Ionization Atmospheric Pressure interface Time-Of-Flight mass spectrometer (CI-APi-TOF, Tofwerk A.G., Switzerland / Aerodyne Research Inc., USA) presented by Junninen et al. (2010) and Jokinen et al. (2012). An Aerodyne High-Resolution Time-of-Flight Aerosol Mass Spectrometer (HR-ToF-

AMS, Aerodyne Research Inc., (DeCarlo et al., 2006)) was deployed to measure real-time, non-refractory particulate matter. In addition, SOA molecular composition with respect to organic acids and high-molecular-weight dimer esters was investigated through off-line filter analysis of the formed α-pinene derived SOA performed using an Ultra High-Performance Liquid Chromatograph coupled to the electrospray ionization source of a Bruker Daltonics quadrupole time-of-flight mass spectrometer (UHPLC/ESI-qToF-MS). Filter sampling was performed once no additional growth in SOA mass was evident from the SMPS and no α-pinene could be detected by the GC-FID. All filter samples were collected by a low volume sampler onto 47 mm 0.20 micrometer PTFE filters (Advantec) situated in stainless steel filter holders at a flow rate of ~27 L min$^{-1}$. After collection, the filter samples were stored at -20 °C until extraction and analysis.

A total of 12 α-pinene oxidation experiments conducted in the AURA smog chamber are presented here (Table 1). Each experiment was performed at temperatures close to either 20 °C, 0 °C or -15 °C to investigate the effect of temperature on the formation and composition of both gas and particle phase organics from dark ozonolysis of α-pinene. Five experiments were conducted with the injection of 10 ppb α-pinene (~0.32 μL, 99%, Sigma Aldrich, Exp. 1.1-1.5) into the ozone-filled chamber (~100 ppb O$_3$). These experiments include oxidation at constant temperatures 20 °C, 0 °C and -15 °C (Exp. 1.1, 1.2 and 1.3) and two temperature ramp experiments (Exp. 1.4 and 1.5). The ramp experiments were performed by injection of α-pinene at a fixed 20 °C (Exp. 1.4) or -15 °C (Exp. 1.5) temperature followed by subsequent ramping to -15 or 20 °C, respectively. In both experiments the gradual and continuous temperature ramping was initiated approximately 40 min after the injection of α-pinene, hence before the SOA mass formation plateaued. In Exp. 1.4 a decrease in temperature from 20 °C to -15 °C was achieved in ~ 100 min, while in Exp. 1.5 heating of the chamber from -15 °C to 20 °C was performed in ~ 140 min. Note that small variations in RH (< 25 %) are observed in between all conducted experiments arising from heating or cooling of the dry chamber air (Table 1).

To investigate the effect of α-pinene concentration on the SOA formation and composition three experiments were performed with 50 ppb of α-pinene (~1.6 μL, 99%, Sigma Aldrich) at 20 °C, 0 °C and -15 °C (Exp. 2.1-2.3) and repeated (Exp. 2.3b and Exp. 3.1-3.3). All experiments were performed without the addition of an OH-scavenger to the chamber.

Throughout this paper SOA density is assumed to be 1 g cm$^{-3}$ as in Kristensen et al. (2017).

## 2.2 Off-line particle analysis by UHPLC/ESI-qTOF-MS

Collected particle filter samples were extracted and analyzed for organic acids and dimer esters. Filter extraction was performed using a 1:1 mixture of methanol and acetonitrile (HPLC grade, Sigma Aldrich). 0.2 μg of camphoric acid recovery standard was added to the filter sample prior to the addition of the extraction mixture to minimize uncertainties related to the off-line extraction and analysis. The samples were placed in a cooled ultrasonic bath for 15 min after which the filters were removed and the extracts were filtered through a Teflon filter (0.45 μm pore size, Chromafil). The extraction solvents were then removed by evaporation over gentle N$_2$ flow and the residues were dissolved by adding 0.2 mL MilliQ water with 10 % acetonitrile and 0.1 % acetic acid. The reconstituted samples were placed in a cooled ultrasonic bath for 5 min and the extract transferred to a HPLC sample vial for analysis. To ensure complete transfer of organics from the evaporated extracts, additional reconstitutions were performed with 0.2 mL MilliQ water with 50 % acetonitrile and 0.1 % acetic acid. Both sample solutions (10 % and 50 % acetonitrile reconstitution) were analyzed by UHPLC/ESI-qTOF-MS immediately after extraction.

The HPLC stationary phase was a Waters Acquity UPLC Ethylene Bridged Hybrid C18 column (2.1 x 100 mm 1.7 um) while the mobile phase consisted of acetic acid 0.1% (v/v) in MilliQ water (eluent A) and acetic acid 0.1% (v/v) in acetonitrile as eluent B. The operating conditions of the mass spectrometer have been described elsewhere (Kristensen and Glasius, 2011). Quantification of the organic acids was performed from eight-level calibration curves (0.1 to 10.0 μg mL$^{-1}$) of the following acids: cis-pinic, cis-pinonic acid, terpenylic acid, diaterpenylic acid acetate (DTAA), 3-methyl-butane tricarboxylic acid (MBTCA). The analytical uncertainty is estimated to be < 20 % for carboxylic acids. Due to lack of authentic standards, the

dimer esters were quantified using DTAA as surrogate standard. DTAA was chosen due to its structural similarities with that of the dimer esters (dicarboxylic acid with ester functionality) as well as similar UHPLC retention time.

## 3. Results and discussions

A representative example of the results obtained from the conducted smog chamber experiments is shown in Fig. 1. Following injection of α-pinene into the ozone-filled chamber at constant temperature and RH, rapid particle formation is captured by the
PSM (measuring particles >1.4 nm) and, shortly after, the SMPS (10-400 nm). In all conducted experiments, maximum particle number concentrations were obtained within 10 minutes after α-pinene injection followed by decay ascribed to wall loss and particle aggregation. Particle number concentrations shown in Fig. 1 (lower panel) are wall-loss corrected. During the course of the experiment, decrease in ozone and α-pinene concentration is accompanied by increase in particle SOA mass concentration as measured by the SMPS. Figure 1 (upper panel) shows wall-loss-corrected SOA mass concentration obtained
from ozone-initiated oxidation of 50 ppb α-pinene (measured by GC-FID). Wall loss corrections were based on first order fits to the SOA mass or number concentration after peak. The grey shaded area represents the time of filter sampling for off-line chemical analysis by UHPLC/ESI-qTOF-MS. Similar figures showing data from all conducted experiments are presented in Supplementary Information.

### 3.1 Particle formation rates and SOA mass yields

Figure 2A-B show the total particle number concentration (# cm$^{-3}$) as measured by the PSM ($D_p$> 1.4 nm ) at 20 °C, 0 °C and -15 °C as a function of time after injection of 10 ppb and 50 ppb α-pinene, respectively. The maximum total particle number concentrations in all experiments are listed in Table 1. Particle formation in terms of number concentrations from the dark ozonolysis of α-pinene at both 10 ppb and 50 ppb VOC concentrations show negative dependence on temperature as has also been reported previously (Jonsson et al., 2008;Kristensen et al., 2017). Maximum particle formation rates (# cm$^{-3}$ s$^{-1}$) are
estimated from linear fits to the experimental time series of total particle number concentration measured by the PSM (Fig. 2A-B). Due to in inhomogeneous mixing in the beginning of the experiments (when the particle formation is most efficient) as well as uncertainties related to the PSM cut-off, the presented particle formation rates should be considered rough estimates and likely not accurate formation rates of e.g. 2 nm particles. However, it is clear that lower temperatures result in relative higher particle formation rates from ozone-initiated oxidation of α-pinene at both low (10 ppb) and high (50 ppb) VOC
concentrations.  At both α-pinene concentrations, a significantly higher particle formation rate is observed when oxidation takes place at 0 °C compared to oxidation performed at 20 °C, with the formation rates being ~ 6 times and ~ 20 times higher at 0 °C than at 20 °C, at 10 ppb and 50 ppb α-pinene experiments, respectively. In comparison, the maximum particle formation rate at both α-pinene concentrations only increased by a factor of 1.5 when performing the oxidation at -15 °C compared to 0 °C.

Figures 2C-D show the evolution of the wall-loss-corrected SOA mass concentration (µg m$^{-3}$) measured by SMPS as a function of time after α-pinene injection. Agreeing with previous studies, SOA mass and mass yields are higher at lower temperatures (Jonsson et al., 2008;Saathoff et al., 2009;Pathak et al., 2007;Kristensen et al., 2017). The reported 18 % and 43 % mass yields at 20 °C and -15 °C, respectively, (50 ppb α-pinene, 100 ppb O$_3$) are in excellent agreement with previous values of 21 % and 39 % from similar oxidation experiments (50 ppb α -pinene, 200 ppb O$_3$) performed under similar conditions in the AURA
chamber (Kristensen et al., 2017). Compared to experiments performed at 20 °C, temperatures of 0 °C and -15 °C respectively result in ~ 50 % and ~ 150 % higher SOA mass concentrations. Interestingly, SOA mass concentration in both 10 ppb and 50 ppb α-pinene oxidation experiments show almost identical response to temperature going from reaction temperatures of 20 °C to -15 °C. In Fig. 2C and 2D the maximum SOA mass is obtained faster at higher reaction temperatures than at lower temperatures. For both low (10 ppb) and high (50 ppb) α-pinene concentrations, maximum SOA mass is reached approximately

140 min, 200 min and 250 min after α-pinene injection at 20 °C, 0 °C and -15 °C, respectively. This is attributed to the faster reaction of α-pinene with ozone at higher temperatures, which is also evident from the calculated α-pinene loss rates shown in the insert in Fig. 2D (50 ppb, GC-FID derived; due to the detection limit of the GC-FID no loss rates could be calculated for 10 ppb α-pinene experiments). Compared to oxidation experiments performed at 20 °C, loss rates of α-pinene were found to be 11 % and 22 % lower at 0 °C and -15 °C reaction temperature, respectively. These results are in good agreement with temperature dependent reaction rates of ozone with α-pinene reported by Atkinson et al. (1982) predicting a 16 (± 4) % and 28 (± 6) % lower reaction rate at 0 °C and -15 °C, respectively, relative to the reaction rate calculated at 20 °C. The discrepancy between measured and literature-derived values is likely due to the presence of OH radicals in the current study. Despite the lower reaction rate for α-pinene ozonolysis at low temperature, the particle number concentrations data, in Fig. 2, show significantly faster particle formation in cold experiments.

**3.2 Particle size distributions and sub-10 nm particles**

Figure 3A shows the particle size distributions as recorded by the SMPS after the maximum SOA mass concentration was reached in (10 ppb) and high (50 ppb) concentration α-pinene oxidation experiments at 20 °C, 0 °C and -15 °C. At both α-pinene concentrations, lower reaction temperatures result in larger particles consistent with the higher SOA masses at 0 °C and -15 °C in Fig. 3C and 3D. The time evolutions of the particle size distribution (Fig. 3B) shows that particles formed at 20 °C are initially larger than the particles formed at the two lower temperatures (0 °C and -15 °C). However, during the course of the experiments, particles formed at the lower temperatures grow more rapidly to larger sizes compared to particles formed at 20 °C. This is likely attributable to enhanced condensation of organics onto the formed particles at lower temperatures.

Figure 4A shows the total particle number concentration (# cm$^{-3}$) obtained by the PSM ($D_p$> 1.4 nm ) and SMPS ($D_p$ in the range 10-400 nm), respectively, as a function of time after α-pinene injection (10 ppb) at both 20 °C and -15 °C. The differences in the total particle number concentrations derived by the two instruments corresponds to the number concentrations in the size range of 1.4 nm to 10 nm. Comparing experiments performed at 20 °C, 0 °C and -15 °C temperatures in Fig. 4B (10 ppb α-pinene), it is clear that the presence of sub-10 nm particles is higher at low reaction temperatures and that the largest temperature dependence is in the range 20 °C to 0 °C.

According to Fig. 4A, sub-10 nm particles constitute ~10 % and 25 % of the total particle number concentration at 20 °C and -15 °C, respectively, even 30 min after α-pinene injection where particle number concentrations have peaked. The size distributions obtained from the SMPS (Fig. 4C), however, indicate very few particles in the smaller size-ranges (i.e. below 20 nm, Fig. 4C insert) at this time of the experiments. A possible explanation for the high sub-10 nm particle number concentration is the presence of a particle distribution mode below the detectable range of the SMPS. Supporting this, Lee et al. (2016) reported an absence of particles in the size range between 3 nm  and 8 nm in ambient air particle measurements (1-600 nm particle size distribution range combining PSM and SMPS data) during the Southern Oxidant and Aerosol Study (SOAS) field campaign. To our knowledge, the current work represents the first indication of this observed "gap" in the particle size distribution in smog chamber experiments, thus emphasizing the need for further studies of sub-10 nm particle formation and detection from VOC oxidation.

**3.3 Molecular composition**

Figure 5 shows the detailed chemical composition of SOA with respect to organic acids and dimer esters at different temperatures (20 °C, 0 °C, and -15 °C) and at two different concentrations of α-pinene (10 ppb and 50 ppb, respectively). The acids and dimer esters identified are similar to those in Kristensen et al. (2017), where suggestions for molecular structures are given. From comparison of repeated experiments (Exp. 2.1 - 2.3 and Exp. 3.1 - 3.3) the uncertainties related to the presented UHPLC/ESI-qTOF-MS results are estimated to be less than 10 %, Figure S3).

Significantly higher particle concentrations of α-pinene-derived organic acids are evident at the lower 0 °C and -15 °C temperatures compared to 20 °C supporting increased condensation of organics at lower temperatures. The total contributions of the identified acids to the formed SOA mass in experiments with an initial α-pinene concentration of 10 ppb are 17 %, 25 % and 32 % at 20 °C, 0 °C and -15 °C, respectively. In comparison, in the 50 ppb α-pinene oxidation experiments, the acids contribute to the formed SOA with 18 %, 38 % and 28 % respectively. For comparison, the fraction of acids was 20 % at 20

°C and 31 % at -15 °C of total SOA mass in experiments with 50 ppb of α-pinene and 200 ppb of ozone (Kristensen et al. 2017). Comparing mass fractions of acids at 20 °C and -15 °C, the mass fraction seems to be highly dependent on temperature and much less dependent on VOC concentration or VOC : $O_3$ ratios.

In all experiments, pinic acid was found to be the dominant identified organic acid, constituting between 5 % and 9 % of the formed SOA mass with highest contributions at lower reaction temperatures. A negative dependence of concentration with

temperature is observed for six of the ten identified organic acids: pinonic acid, pinalic acid, oxopinonic acid, OH-pinonic acid, pinic acid and terpenylic acid. Diaterpenylic acid acetate (DTAA), diaterpenylic acid (DTA) and 3-methyl-butanetricarboxylic acid (MBTCA), however, show lower concentrations at the lower temperatures compared to at 20 °C. The effect of temperature seems to correlate with the O:C ratio (i.e. degree of oxidation of the organic acids) as shown in Fig. 5. Acids with lower O:C ratios (O:C < 0.4) are found at higher concentrations at lower temperatures (0 °C and -15 °C). This can

likely be attributed to an enhanced condensation of these species onto the SOA particles at temperatures of 0 °C and -15 °C. In contrast, although contributing significantly less to the SOA mass, the more oxidized compounds (O:C > 0.4, DTAA, DTA and MBTCA) show an opposite trend with respect to temperature (Fig. S3). As MBTCA, DTAA and DTA are hypothesized to be formed from gas-phase oxidations involving OH-radicals (Szmigielski et al., 2007;Vereecken et al., 2007;Müller et al., 2012;Kristensen et al., 2014), their lower concentrations at lower reaction temperatures could be explained by (1) changes in

reaction pathways or branching of the intermediates of MBTCA, DTAA and DTA, (2) lower gas-phase concentration of the first generation of oxidized organic compounds due to higher degree of condensation (e.g. MBTCA is formed from gas-phase oxidation of pinonic acid), and (3) reduction of OH radical production and hence reduced oxidation by OH-radicals at lower reaction temperatures (Jonsson et al., 2008;Tillmann et al., 2010). The former (1) is supported by Müller et al. (2012) and Donahue et al. (2012a) reporting lower particle-phase MBTCA correlating with decreased pinonic acid vapor fractions at lower

temperatures. Both explanations are supported by lower O:C ratios of the total α-pinene-derived SOA formed at the lower temperatures as reported in Jensen et al. (2020).

A total of 30 dimer esters were identified in the collected SOA particle samples from both 10 ppb and 50 ppb α-pinene ozonolysis experiments. In the 10 ppb α-pinene experiments, total concentrations of dimer esters of 0.64, 0.51 and 0.46 µg m$^{-3}$ were found in SOA particles formed at 20 °C, 0 °C and -15 °C, respectively, thus showing a small positive dependence on

temperature. The corresponding mass fractions were 11.1 %, 7.5 % and 3.8 %. For particles formed from higher 50 ppb α-pinene concentrations, the temperature dependence is less obvious with total dimer ester concentration of 3.1, 4.3, and 3.3 µg m$^{-3}$ at 20, 0, and -15 °C, respectively. The corresponding mass fractions of the dimer esters are: 8.9 %, 8.0 %, and 3.9 %, thus similar to fractions reported for the 10 ppb α-pinene oxidation experiments, thus, as in the case of the organic acids, the mass fraction of dimer esters seems to be highly dependent on temperature and much less dependent on VOC concentrations or

VOC:O3 ratios. In relation, Kourtchev et al. (2016) observed a positive relationship between temperature and oligomer fraction in aerosol samples collected at Hyytiälä in summer 2011 and 2014 but ascribed this to differences in the VOC emissions. However, the current study, indicates that temperature alone may influence the aerosol fraction of dimeric compounds thus supporting to the ambient observation in Kourtchev et al. (2016).

Looking at the concentrations of the 30 individual dimer esters identified in the current work, their formation is affected differently by the reaction temperature (Fig. 5). Figure 6A shows the yields of the identified dimer esters at -15 ° C relative to 20 °C (yield (-15 °C) / yield (20 °C)) as a function of the number of carbon atoms in the dimer esters. From this, it is clear that

the effect of temperature on the formation of individual dimer esters is compound specific. Of the 30 dimer esters, increased particle concentrations at lower temperatures is most often related to species with high carbon number (i.e. $C_{19}$ species). Also, Fig. 6A shows that the temperature effect on the dimer ester concentration to a large extent depends on the O:C ratio (indicated by color scale) of the individual species. This is highlighted also in Fig. 6B showing a more significant decrease in relative yields of dimer esters with higher oxygen number (i.e. more oxidized) at the lower -15 °C temperature in both 10 ppb and 50 ppb α-pinene experiments. Accordingly, the dimer esters are grouped based on their O:C ratios, with the more oxidized dimer esters having an O:C > 0.4; the O:C value above which all dimer esters show decreased concentration at the lower -15 °C compared to 20 °C (Figure S4). The total concentrations of the low (< 0.4) and high (> 0.4) O:C dimer esters are shown in Fig. 5B and 5D inserts.

## 3.4 Temperature ramping

Normann Jensen et al. (2020) present a detailed analysis on changes in AMS derived elemental chemical composition during temperature ramps in the ACCHA campaign and show that temperature at which the particles are formed to a large extend determines the elemental composition also after subsequent heating or cooling by 35 °C. During two temperature ramp experiments, we also studied how the concentration, mass fraction and molar yields of carboxylic acids and dimer esters as identified in the off-line analysis varied. The experiments involving temperature ramps were conducted to examine whether the observed effect of temperature on the dimer ester concentration is attributed to gas-to-particle partitioning or stabilization/decomposition of the hydroperoxide-containing species.

Figure 7A shows the SOA mass formation (wall-loss corrected) from the 10 ppb α-pinene oxidation experiments performed at constant 20 °C (Exp. 1.1) and -15 °C (Exp. 1.3) and from the two temperature ramping experiments (Exp. 1.4 and 1.5) in which the chamber temperature was ramped up or down ~ 40 min after injection of α-pinene. In the ~ 40 min following the injection of α-pinene, SOA mass in both ramp experiments show good agreement with the constant temperature experiments performed at similar starting temperature (Exp. 1.1 and 1.3). Likewise, particle formation rates and maximum particle number concentrations obtained in Exp. 1.4 and 1.5 prior to temperature changes (Table 1) also resemble those of Exp. 1.1 and 1.3. Thus the initial evolution of the particle formation in Exp. 1.1 (constant 20 °C) and Exp. 1.4 (ramp experiment initiated at 20 °C) and in Exp. 1.3 (constant -15°C) and Exp. 1.5 (ramp experiment initiated at -15 °C) are comparable.

From Fig. 7A, cooling and heating the chamber imply a quasi-immediate effect on the SOA mass concentration. In Exp. 1.4, decreasing the temperature results in a maximum SOA mass concentration being ~ 30 % higher than the maximum SOA mass in Exp. 1.1 performed at a constant 20 °C (8 versus 6 µg m$^{-3}$). Reversely, heating the chamber from -15 °C to 20 °C (Exp. 1.5) results in a ~ 30 % lower SOA mass (10 versus 15 µg m$^{-3}$) compared to SOA formed and kept at -15 °C (Exp. 1.3). The changes in particle size distributions as a result of temperature ramping (Fig. 7B) are attributed to evaporation during heating and condensation during cooling. Interestingly, despite the loss or gain of particle mass during temperature ramping the SOA mass concentration is closer to the experiments performed at the initial temperature rather than the final temperature even after a temperature ramp of 35 °C. This, in turn, suggests that the formation conditions to a large extent govern the type of SOA that was formed, rather than the conditions the SOA was later exposed to.

Figure 8 shows the mass concentrations, mass fractions and molar yields of organic acids and dimer esters in SOA particles from constant temperature oxidation experiments at 20 °C and -15 °C (Exp. 1.1 and 1.3, respectively) and from the two temperature ramp experiments (Exp. 1.4 and 1.5). Upon heating SOA particles from -15 °C to 20 °C, a loss of particle mass (5 µg m$^{-3}$, Fig. 7) was observed and is attributed to evaporation of organics. This is supported by a 2.8 µg m$^{-3}$ lower organic acid concentration in particles exposed to heating in Exp. 1.5 compared to particles formed at the constant -15 °C (Exp. 1.3), accounting for ~ 60 % of the evaporated organics. These results underline the semi-volatile nature of the identified organic acids (i.e. pinalic acid, pinonic acid, oxopinonic acid, OH-pinonic acid, pinic acid, terpenylic acid and terebic acid) and show that these acids are readily removed from organic aerosols during heating thus explaining previously reported reduced aerosol

mass fraction from heating α-pinene SOA (Pathak et al., 2007). In contrast, cooling of the formed SOA particles in Exp.1.4 resulted only in a small 0.3 μg m$^{-3}$ increase in organic acid concentration making up ~ 15 % of the reported 2 μg m$^{-3}$ increase in SOA mass compared to the constant 20 °C experiment. These results indicate limited gas-to-particle phase condensation of the semi-volatile organic acids upon cooling the SOA. The current findings are in agreement with that of Zhao et al. (2019a) who found that during cooling of α-pinene SOA particles the resulting mass growth was largely over predicted by the applied volatility basis set model. As suggested in Zhao et al. (2019a) the observed limited condensation of semi-volatiles to SOA particles during cooling is likely attributed to an expected high particle viscosity at the relative low RH conditions of the performed oxidation experiments. Under these conditions, condensed SVOCs are likely confined at the near-particle-surface region thus impeding further partitioning of the gas-phase SVOCs (Renbaum-Wolff et al., 2013). In the current study, this is supported by the relatively small increase in organic acids (Fig. 8) as well as almost negligible increase in particle size (Fig. 7B) observed upon cooling the SOA particles from 20 °C to -15 °C. Also, if SVOCs reside near the particle surface, this would indeed explain the effective evaporation of these compounds (i.e. the identified organic acids, Fig. 8) and subsequent reduction in particle size (Fig. 7B) observed in Exp. 1.5.

With respect to dimer esters, thermodynamics, or partitioning, could be considered as a possible explanation for the observed temperature responses of the high and low O:C dimer esters. As reported in Kristensen et al. (2017), the identified dimer esters span across a wide range of volatilities. Here, many of the low O:C dimer esters may be sufficiently volatile to allow considerable fractions to exist in the gas phase at high temperature. Consequently, the increased particle-phase concentration observed at the lower temperatures may solely be attributed to enhanced gas-to-particle phase partitioning of these species. Supporting this, Mohr et al. (2017) identified dimeric monoterpene oxidation products ($C_{16-20}H_yO_{6-9}$) in both particle phase and gas phases in ambient air measurements in the boreal forest in Finland. However, Figure 8 reveals that heating and cooling of α-pinene-derived SOA particles, after their formation, results in insignificant or small changes in the concentrations, mass fractions, and molar yields of dimer esters when compared to experiments performed at constant temperatures. This indicates that the dimer esters are not subjected to evaporation or decomposition within the studied temperature range and timeframe. Also, the results indicates that the SOA particle dimer ester concentration is, within the timeframe of the performed experiments, unaffected by any changes to the particle phase-state associated with either the performed cooling or heating of the particles. This could thus suggest that the formation of dimer esters proceed through gas-phase mechanism rather than reactions in the particle phase. Thus, the results show that the concentration of the dimer esters in SOA from α-pinene oxidation is largely determined by the temperature at which the SOA is formed rather than subsequent exposure to higher or lower temperatures.

**3.5 Dimer esters formation**

Gas-phase formation of dimer esters from α-pinene ozonolysis has been suggested to proceed through the reaction of stabilized Criegee Intermediate (sCI) with carboxylic acids resulting in the formation of a class of ester hydroperoxides, α-acyloxyalkyl hydroperoxides (α-AAHPs) (Zhao et al., 2018a;Kristensen et al., 2016). In relation, Claflin et al. (2018) stated that the only known gas-phase mechanism for forming esters in their experiments was the reaction of sCI with carboxylic acids under dry condition. As the reactions of sCI are expected to be temperature-dependent this could explain the lower formation of some of the identified dimer esters (the high O:C dimers) in the current study. Also, as suggested by Kristensen et al. (2017), temperature-modulated condensation of the gas-phase carboxylic acid precursors may also contribute to the observed temperature effects on the formation of dimer esters. As the condensation of the carboxylic acids is vapor pressure dependent it is expected that the less volatile (i.e. more oxidized) species are more effectively depleted from the gas phase at lower temperatures hence hindering their gas-phase reactions with the sCI and reducing formation of the more oxidized dimer esters. Consequently, at the lower temperatures the formation of dimer esters is more likely to proceed through reactions of sCI with the more volatile and less oxidized carboxylic acids species thus potentially explaining the observed increased formation of

the low O:C dimer esters in the 0 °C and -15 °C experiments. In addition to the observed temperature dependence of the low O:C dimer esters an increased stability of hydroperoxide-containing species in SOA particles is expected at lower temperatures (Zhao et al., 2018a).

While the formation of α-AAHPs through sCI + carboxylic acids is a plausible mechanism related to many of the dimer esters identified in the current study, this may not be true for all species. In particular, studies on the MW 368 dimer ester (pinonyl-pinyl ester, $C_{19}H_{28}O_7$) and MW 358 (pinyl-diaterpenyl ester, $C_{17}H_{26}O_8$) conclude that the chemical structures of these esters do not include hydroperoxide functionalities (Beck and Hoffmann, 2016;Kahnt et al., 2018). Consequently, Kahnt et al. (2018) recently suggested an alternative mechanism for the formation of these species involving gas-phase formation and subsequent

rearrangement of unstable $C_{19}H_{28}O_{11}$ HOM species formed from RO$_2$ + R'O → RO$_3$R reaction of an acyl peroxy radical and an alkoxy radical. Specifically, Kahnt et al. (2018) explains that the $C_{19}H_{28}O_{11}$ HOM species decompose through the loss of oxygen or ketene resulting in the formation of the MW 368 dimer ester (pinonyl-pinyl ester, $C_{19}H_{28}O_7$) and the MW 358 dimer ester (pinyl-diaterpenyl ester, $C_{17}H_{26}O_8$), respectively. Interestingly, in the current study, the particle-phase concentration of the MW 368 dimer ester increases at lower temperatures, while the opposite is seen for the MW 358 dimer esters (Fig. 5), thus

indicating significant differences in the mechanism responsible for the formation of these particular dimer esters. In accordance with the mechanism suggested by Kahnt et al. (2018), the different responses to temperature of the two dimer esters could be explained by (1) a temperature modulated formation of the alkoxy radicals related to the 5- and 7-hydroxy-pinonic acid involved in the formation of the two $C_{19}H_{28}O_{11}$ HOM species decomposing to the MW 368 and MW 358 dimer esters, respectively; or (2) temperature-dependent decomposition and rearrangement of the $C_{19}H_{28}O_{11}$ HOM species suppressing the

more complex decomposition and rearrangement mechanism in which carbon-containing entities (e.g. ketene) is expelled from the HOM. However, these explanations are not supported by the temperature ramping experiments performed herein indicating that the formation of the dimer esters is determined by the initial reaction temperature and remains relatively unaffected by heating or cooling. In addition, although accounting for the differences in the temperature response of the MW 368 and MW 358 dimer esters, and to some extent the increased particle concentration of many higher carbon number dimer esters (i.e. C$_{19}$

species, Fig 6A), a temperature-modulated decomposition does not solely explain the observed decrease in relative yields of dimer esters with higher oxygen number (Fig. 6B). To explain this, the formation and subsequent decomposition of more oxidized HOMs need to be considered as possible precursors for the more oxygen-rich dimer esters.

A detailed study on the formation of HOMs during the ACCHA campaign is presented by Quéléver et al. (2019). Here, significantly lower (by orders of magnitude) HOM gas-phase concentrations are observed in -15 °C experiments compared to

20 °C experiments. As the HOMs form through autoxidation of RO$_2$, low temperatures and thus decreased autoxidation is expected to result in lower formation of HOMs with reduced formation of the more oxygenated species. Interestingly, however, Quéléver et al. (2019) observed no correlation between the degree of oxidation (i.e. O:C-ratio) of the identified HOMs and the magnitude by which the formation of these was reduced at lower temperatures. This is in contrast to the observed temperature effects on the formation of the identified dimer esters presented in the current study (Fig. 7), and thus rule out the formation

and decomposition of more oxidized HOMs as mechanism for the formation of more oxygen-rich dimer esters. In Quéléver et al. (2019), one possible interpretation of the observed temperature effect on HOMs is that the rate-limiting step in the autoxidation chain takes place already in the first steps of autoxidation. This is supported by the observed decreased concentration of dimer esters with a higher number of oxygen atoms, which also indicates that the formation of the identified dimer esters could proceeds through reaction of products from RO$_2$ autoxidation. The involved oxidation state of these products

may vary depending on the degree of autoxidation undertaken by the RO$_2$ radical as well as the radical termination of these including unimolecular processes leading to loss of OH or HO$_2$ or bimolecular reactions with NO, HO$_2$ or other RO$_2$ resulting in the formation of ROOR dimers. As the autoxidation as well as the bimolecular reactions of RO$_2$ radicals are temperature-dependent, these processes may provide explanation for the observed response to temperature of the different dimer esters. Claflin et al. (2018) showed that the autoxidation and radical termination reactions of the Criegee Intermediate RO$_2$ radicals

may results in a plethora of different products covering a range of oxidation states and functionalities; including multifunctional $RO_2$ radicals, hydroperoxide, carbonyl, alcohol, carboxylic and peroxycarboxylic acid, dialkyl and diacyl peroxides. Of these, carbonyls, alcohols and carboxylic acids reacts readily with sCI resulting in dimeric compounds such as secondary ozonides, α-alkoxyalkyl hydroperoxides (AAAHs), and α-AAHP, respectively (Chhantyal-Pun et al., 2018;McGillen et al., 2017;Khan et al., 2018;Claflin et al., 2018). In relation, $RO_2$–$RO_2$ reactions have been proposed as conceivable mechanism for the

formation of dimers from α-pinene ozonolysis (Claflin et al., 2018;Zhao et al., 2018b). Here, $RO_2$ produced from the isomerization or decomposition of Criegee Intermediates are suggested to participate in $RO_2$-$RO_2$ reactions resulting in dialkyl or diacyl peroxides. The formation of dimer esters through reactions of $RO_2$ is supported by an observed decrease in dimer esters concentrations at higher levels of NOx in ambient air measurement in Hyytiälä, Finland (Kristensen et al., 2016) and supports the formation proposed by several studies (Ehn et al., 2014;Berndt et al., 2018;Zhao et al., 2018b) involving $RO_2$

cross-reactions as likely route of gaseous dimer formation. The $RO_2$–$RO_2$ reaction is expected to compete with the reactions of $RO_2$ with $HO_2$ radicals. In the absence of an OH-scavenger, the performed oxidation experiments will include the formation of OH-radicals from the gas-phase reaction of α-pinene with $O_3$. The formed OH-radicals reacts readily with $O_3$ yielding $HO_2$-radicals available for $RO_2$-$HO_2$ reactions. As both reactions ($O_3$ + α-pinene and $O_3$ + OH) have a positive temperature dependence, the formation of $HO_2$ and its subsequent reaction with $RO_2$ is expected to increase at the higher reaction

temperatures. In Simon et al. (2020) the higher concentration of $HO_2$ leads to an increased competition with the $RO_2$-$RO_2$ self-reaction, which reduced the formation of HOM dimers but increased HOM monomers. However, in the current study, reduction in the concentration of dimer esters due to increased $RO_2$-$HO_2$ competition at higher temperatures is only observed in the case of the low O:C dimer esters, such as the pinonyl-pinyl ester. In the case of the higher O:C dimers, it appears that a suppressed competition of $HO_2$ with $RO_2$ at the lower temperatures is less important, and that other mechanisms are responsible for

observed temperature effects on the more oxidized dimers.

Recently, heterogeneous chemistry has been suggested as a possible pathway for the formation of dimer ester in β-pinene oxidation experiments. Here, dimer esters are suggested to form from semivolatile dicarboxylic acids (e.g. pinic acid) undergoing traditional equilibrium gas-particle partitioning with subsequent reactive uptake of the gas-phase, OH-derived monomers on collision with particle surfaces to form dimer esters (Kenseth et al., 2018). In Kenseth et al. (2018) the suggested

mechanism is supported by an observed increased formation of certain dimer esters in oxidation experiments including seed particles enriched with pinic acid as well as an observed suppressed formation of some dimer esters in the presence of an OH-scavenger. In relation, Zhao et al., (2019b) recently reported dimer formation through dimerization by organic radical (i.e., peroxy, $RO_2$, and alkoxy radicals, RO) cross reactions during heterogeneous OH-initiated oxidative aging of oxygenated organic aerosol.

In the current study, heterogeneous chemistry and OH-dependency could help explain the observed response to temperature of the identified dimer esters. At the lower 0 and -15 °C reaction temperatures increased condensation of semivolatile organic acids, such as pinic acid (Fig. 5 & Fig. 7), could facilitate the formation of dimer esters from these monomeric species. In addition, reduced OH radical production at lower reaction temperatures could explain the suppressed formation of the higher O:C dimer esters from reduced formation of more oxidized organic species, such as MBTCA and DTAA (Fig. S3), and other

OH-depended monomers available for reactive uptake unto the formed SOA particles or gas-phase reactions. This is somewhat supported by an observed correlations between the particle phase concentration of pinic acid and the pinonyl-pinyl ester ($R^2$ = 0.92, Fig. S6) whilst the more oxidized dimer ester pinyl-diaterpenyl ester correlate better with MBTCA ($R^2$ = 0.99, Fig. S6) rather than pinic acid ($R^2$ = 0.02). Although correlation between particle-phase monomeric organic acids and dimer esters could suggest purely particle-phase chemistry as mechanism for dimer ester formation this is not supported by the relatively

small changes in the dimer ester concentration following heating or cooling of the formed SOA particles (Fig. 8). In particular, temperature-ramping experiments show that while the particle-phase concentrations of pinic acid and MBTCA are affected by both heating and cooling of the SOA particles, both pinonyl-pinyl ester and pinyl-diaterpenyl ester are unaffected with

concentrations determined by the initial reaction temperatures (Fig. S6). The observed temperature effects on dimer ester formation could be ascribed to oxidation by OH-radicals of the dimer ester precursors, whether formed from autoxidation processes or $RO_2$ termination pathways, or of the dimer esters themselves thus leading to the formation of high O:C dimer esters.

We propose that, although different in chemical structures and O:C-ratios, dimer esters and HOMs may be linked via their formation mechanisms, both involving $RO_2$. The particle-phase dimer esters and the gas-phase HOMs may merely represent two different fates of the $RO_2$ radicals. If conditions are favorable and efficient autoxidation takes place, this will result in the formation of HOMs, which by the definition recommended by Bianchi et al. (2019) in this case means any molecule with 6 or more oxygen atoms that has undergone autoxidation. On the other hand, dimer esters could be the product of cross reactions of $RO_2$ formed from autoxidation or formed from OH-oxidation of the autoxidation termination products with O:C ratios influenced by the number of potential autoxidation steps undertaken by the involved $RO_2$ species prior to reaction or termination. Whether the formation of dimer esters proceeds through gas-phase or heterogeneous $RO_2$-$RO_2$ cross reactions of species formed from autoxidation or from OH-oxidation of termination products or through monomeric compounds reacting with sCI is yet to be determined.

## 4. Conclusions

The formation of SOA from dark ozonolysis of α-pinene is highly influenced by temperature. At sub-zero temperatures, such as -15°C, more effective nucleation gives rise to significantly higher particle number concentration compared to similar experiments performed at 20 °C, where the vast majority of laboratory studies are conducted. In addition, the SOA mass concentration resulting from α-pinene ozonolysis shows a strong temperature dependence attributed to increased condensation of semi-volatile oxidation products at lower temperatures. This is supported by higher concentration of semi-volatile organic acids, such as pinic acid and pinonic acid, in SOA particles formed at 0 °C and -15 °C compared to particles formed at 20 °C. In addition to organic acids, the contribution of high-molecular-weight dimer esters to the formed SOA is also affected by temperature. Underlining the chemical complexity of α-pinene SOA, the 30 quantified dimer esters showed different behaviors with respect to temperature, with the suppressed formation of the more oxidized dimer esters (O:C > 0.4) at low reaction temperatures. This feature is not seen in the case of the least oxidized dimer esters (O:C < 0.4), showing a small increase in particle concentration at lower temperatures. Similar to the high O:C dimer ester compounds, α-pinene ozonolysis experiments performed at lower temperatures result in lower HOM formation from autoxidation in the gas-phase. We suggest that the identified dimer esters may form through $RO_2$ cross-reactions with the oxidative state (i.e. O:C ratio) of the resulting dimer esters highly influenced by available OH-radicals for the formation of more oxidized organic radicals prior to dimerization. Consequently, we speculate that the observed temperature effects on the identified dimer esters can be impacted by the OH-formation and reactivity at the studied temperatures. These results indicate that temperature not only affects the formation of SOA mass in the atmosphere but also alters the chemical composition through condensation and evaporation of semi-volatile species, changes in the formation of HOMs and finally in the reaction pathways leading to the formation of dimer esters having high and low O:C-ratios.With respect to dimer esters, no decomposition or evidence of partitioning changes between the aerosol and the gas phase were observed from heating or cooling the SOA particles, suggesting that (1) the formation and, consequently, concentration of dimer esters is dictated by the VOC oxidation conditions and (2) once formed, dimer esters remain in the particle phase, representing a core compound within SOA and thus an efficient organic carbon binder to the aerosol phase. In relation, the presented results from temperature ramping show that final SOA mass concentration obtained from dark ozonolysis of α-pinene is more dependent on the initial reaction temperatures rather than temperatures to which the formed SOA is subsequently exposed. In conclusion, this means that the changes in ambient temperatures in areas in which emissions and oxidation of VOCs, such as α-pinene, is likely to result in significant changes to the resulting SOA mass as well

as the chemical composition of the SOA. As global temperatures are expected to rise, especially in the Nordic regions of the boreal forests, this means that although less SOA mass is expected to form from the oxidation of emitted biogenic VOCs, the temperature-induced changes to the chemical composition may result in more temperature resistant SOA influencing cloud-forming abilities from increased content of oligomeric compounds (i.e. dimer esters).

**Author contributions**

MB, ME, MG and HBP supervised the ACCHA campaign. KK, LLJQ, SC, ME, and MB designed the experiments. KK, LNJ, SC initialized the chamber for experiments. KK and LNJ measured and analyzed the aerosol phase. KK, BR, and RT measured and analyzed the VOCs and their oxidation production. LLJQ performed the measurement and analyzed the gas-phase HOMs. JE guided and helped with the interpretation of the dimer ester data and formation pathways. KK prepared the manuscript with the contributions from all co-authors.

**Acknowledgement**

We thank Aarhus University and Aarhus University Research Foundation for support. This work was supported by the European Research Council (Grant 638703-COALA) and the Academy of Finland (grants 307331, 317380 and 320094). KK acknowledge the Carlsberg Foundation (Grant CF18-0883) for financial support. JE thanks the Villum foundation and the Swedish Research Council Formas project number 2018-01745-COBACCA for financial support.

**Table 1.** Overview of conducted α-pinene ozonolysis experiments

| ID | Date | Exp. Type | α-pinene (ppb) | Ozone (at injection) (ppb) | Temp.[a] (at injection) (°C) | RH[a] (at injection) (%) | Temp. avg.[b] (°C) | RH avg.[b] (%) | VOC loss rate[c] ($h^{-1}$) | Max SOA mass[d] ($\mu g\ m^{-3}$) | Max particle number[d] (10-400nm) (# $cm^{-3}$) | Max particle number (>1.4nm) (# $cm^{-3}$)[e] |
|---|---|---|---|---|---|---|---|---|---|---|---|---|
| 1.1 | 20161202 | Low load, 20 °C | 10 | 104 | 20.2 | 0 | 20.3 (±0.1) | 0.8 (±0.8) | N/A | 6 | $3.9 \cdot 10^4$ | $5.2 \cdot 10^4$ |
| 1.2 | 20161208 | Low load, 0 °C | 10 | 105 | 0.9 | 2.9 | 0.7 (±2.7) | 7.1 (±4.1) | N/A | 9 | $7.9 \cdot 10^4$ | $17.0 \cdot 10^4$ |
| 1.3 | 20161207 | Low load, -15 °C | 10 | 106 | -13.7 | 8 | -14.7 (±0.1) | 12.9 (±5.3) | N/A | 15 | $7.4 \cdot 10^4$ | $19.5 \cdot 10^4$ |
| 1.4 | 20161209 | Low load, 20 to -15 °C | 10 | 103 | 19.9 | 0 | | N/A | N/A | 8 | $4.4 \cdot 10^4$ | $8.4 \cdot 10^4$ |
| 1.5 | 20161220 | Low load, -15 to 20 °C | 10 | 113 | -14 | 11.7 | | 5.3 (±4.0) | N/A | 10 | $9.4 \cdot 10^4$ | $20.0 \cdot 10^4$ |
| 2.1∞ | 20161212 | High load, 20 °C | 50 | 105 | 19.8 | 0 | 20 (±1.1) | 1.1 (±1.3) | 1.0 | 50 | $8.2 \cdot 10^4$ | $11.0 \cdot 10^4$ |
| 2.2 | 20161219 | High load, 0 °C | 50 | 107 | -0.4 | 6.9 | -0.3 (±0.1) | 6.9 (±0.8) | 0.9 | 65 | $19.0 \cdot 10^4$ | $>70.0 \cdot 10^4$ * |
| 2.3a | 20161213 | High load, -15 °C | 50 | 101 | -13.1 | 6.7 | -14.9 (±0.6) | 11.9 (±4.3) | 0.8 | 120 | $24.0 \cdot 10^4$ | $>70.0 \cdot 10^4$ * |
| 2.3b∞ | 20161221 | High load, -15 °C | 50 | 113 | -14.0 | 19.8 | -15.0 (±0.2) | 24.7 (±3.6) | 0.8 | 131 | $16.0 \cdot 10^4$ | $51.2 \cdot 10^4$ |
| 3.1 | 20170112 | High load, 20 °C | 50 | 100 | 20.3 | 0 | 20.1 (±0.5) | 2.4 (±2.0) | 1.1 | 50 | $7.8 \cdot 10^4$ | $8.4 \cdot 10^4$ |
| 3.2 | 20170116 | High load, 0 °C | 50 | 105 | 0.2 | 8.6 | 0.0 (±0.1) | 8.7 (±1.1) | 1.0 | 78 | $25.0 \cdot 10^4$ | $>70.0 \cdot 10^4$ * |
| 3.3 | 20170113 | High load, -15 °C | 50 | 105 | -14.5 | 11.2 | -14.8 (±0.6) | 15.8 (±4.5) | 0.9 | 115 | $23.0 \cdot 10^4$ | $>70.0 \cdot 10^4$ * |

[a] Temperature and RH measured in centre of Teflon bag,

[b] Average temperature and RH (±std.dev.) over the entire experiment,

[c] VOC loss rates are estimated from GC-FID measurements

[d] Measured by SMPS (10-400nm), [e] Measured by PSM (>1.4nm),

* PSM overloaded

∞ Temperature ramps were performed after the stable temperature phase and is presented in Jensen et al., (2020).

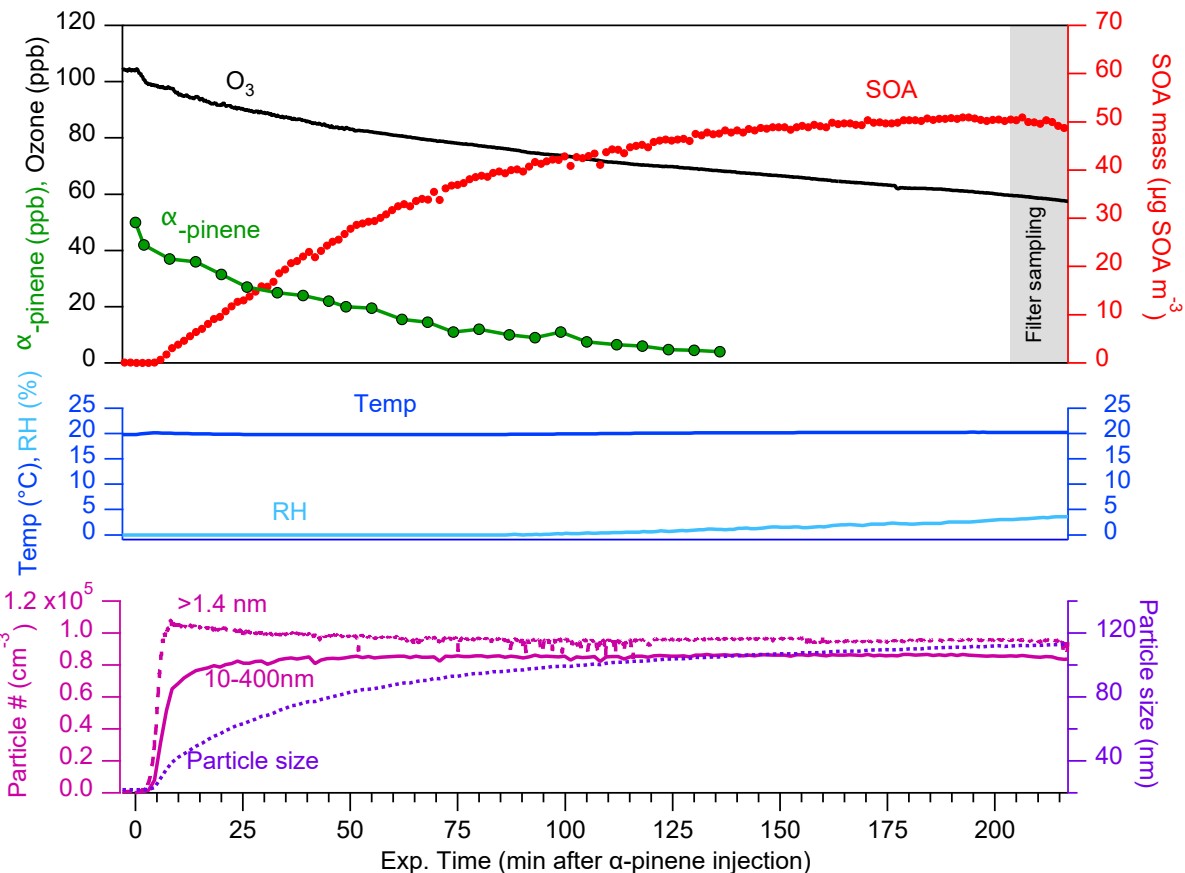

**Figure 1**. Concentration of $O_3$ (ppb, black), α-pinene (ppb, green), wall-loss corrected SOA mass (µg m-3, red) and particle number concentration (# cm-3, measured by PSM (>1.4nm) and SMPS (10-400nm), dark red) and the geometric mean particle size (nm, by SMPS, violet) along with recorded RH (%, teal) and temperature (°C, blue) during a 50 ppb α-pinene oxidation experiment performed at 20 °C (Exp. 2.1).

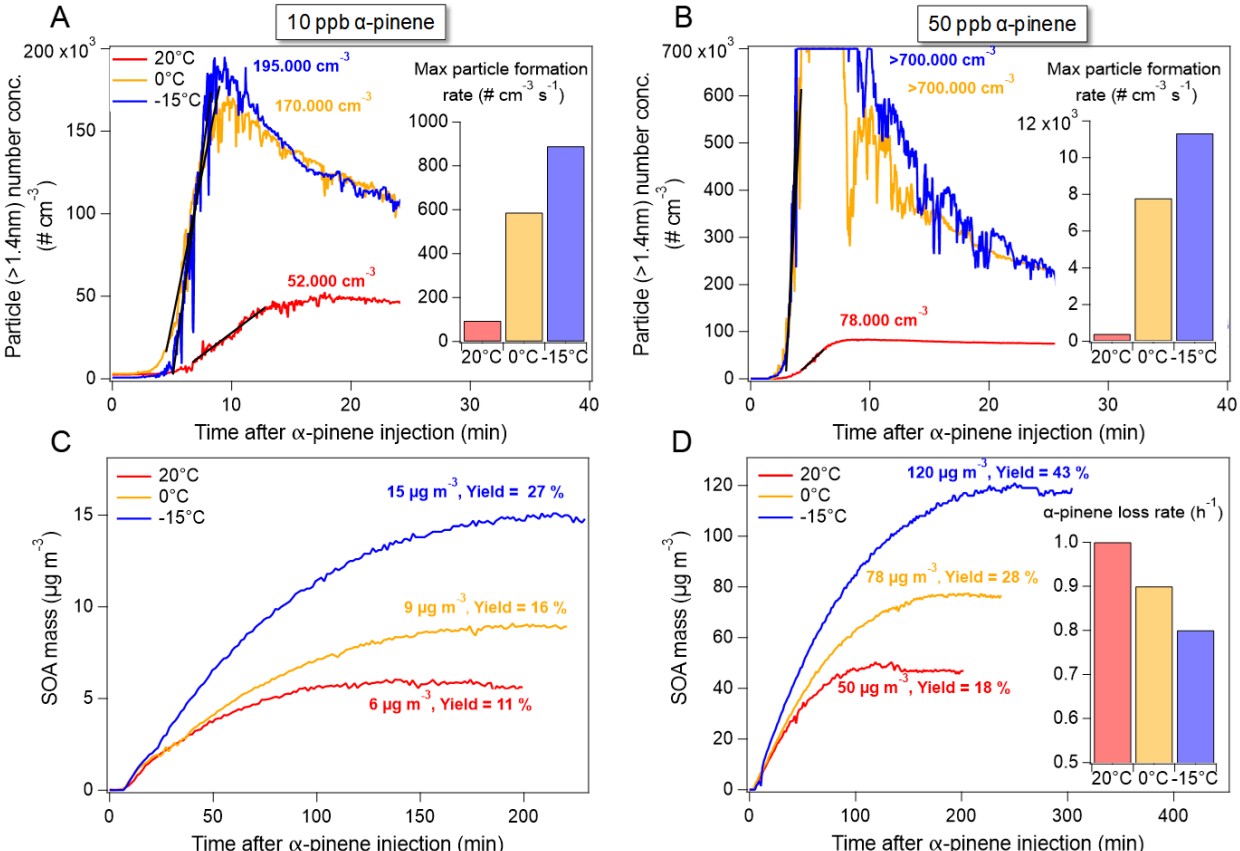

**Figure 2**. Effect of reaction temperature and initial VOC concentration on SOA formation with respect to particle number concentration (A, B) and wall-loss-corrected SOA mass concentration (C, D) at α-pinene concentrations of 10 ppb (A, C) and 50 ppb (B, D). Inserts show particle formation rates (# cm$^{-3}$ s$^{-1}$) as estimated from linear fits to the experimental data (A, B). The insert in D shows the loss rate of α-pinene (h$^{-1}$) at different temperatures as derived from an exponential fit to the measured concentration of α-pinene.

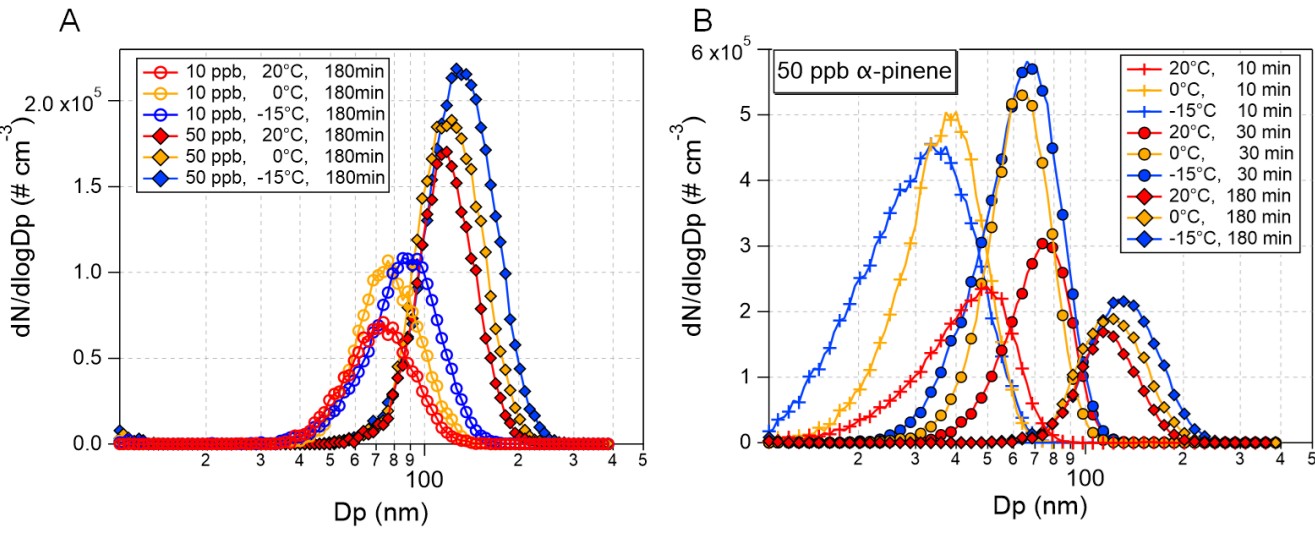

**Figure 3**. **A)** Particle size distribution recorded 180 minutes after the ozone-initiated oxidation of 10 (Exp. 1.1-1.3) and 50 ppb (Exp. 2.1-2.3) α-pinene at 20 °C (red), 0 °C (orange), and -15 °C (blue). **B)** Particle size distribution recorded 10, 30 and 180 minutes after injection of 50 ppb α-pinene at 20 °C (Exp. 2.1, red), 0 °C (Exp. 2.2, orange), and -15 °C (Exp. 2.3, blue).

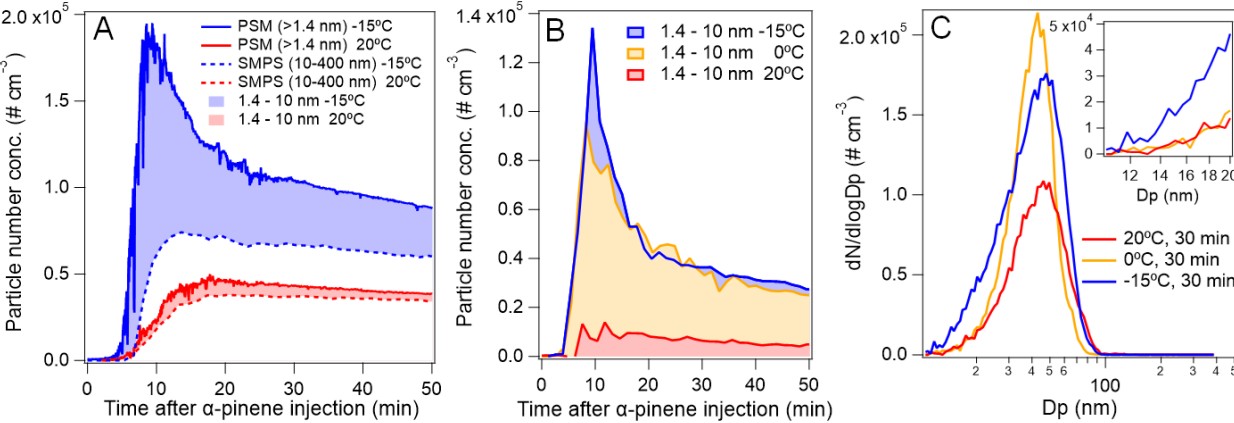

**Figure 4**. **A)** Effect of temperature on particle number concentrations (# cm⁻³) as derived from the SMPS (10-400 nm particle size range, broken line) and PSM (>1.4 nm particle size range, solid line) in experiments with an initial α-pinene concentration of 10 ppb. (Exp. 1.1, red, and Exp. 1.3, blue). **B)** Particle number concentrations (# cm⁻³) of 1.4 – 10 nm particles in experiments with an initial α-pinene concentration of 10 ppb performed at -15°, 0 °C and 20 °C (Exp. 1.1, 1.2, and 1.3, respectively). **C)** Particle size distributions recorded 30 minutes after the ozone-initiated oxidation of 10 ppb α-pinene performed at 20 °C (red), 0 °C (orange), and -15 °C (blue) (Exp. 1.1-1.3).

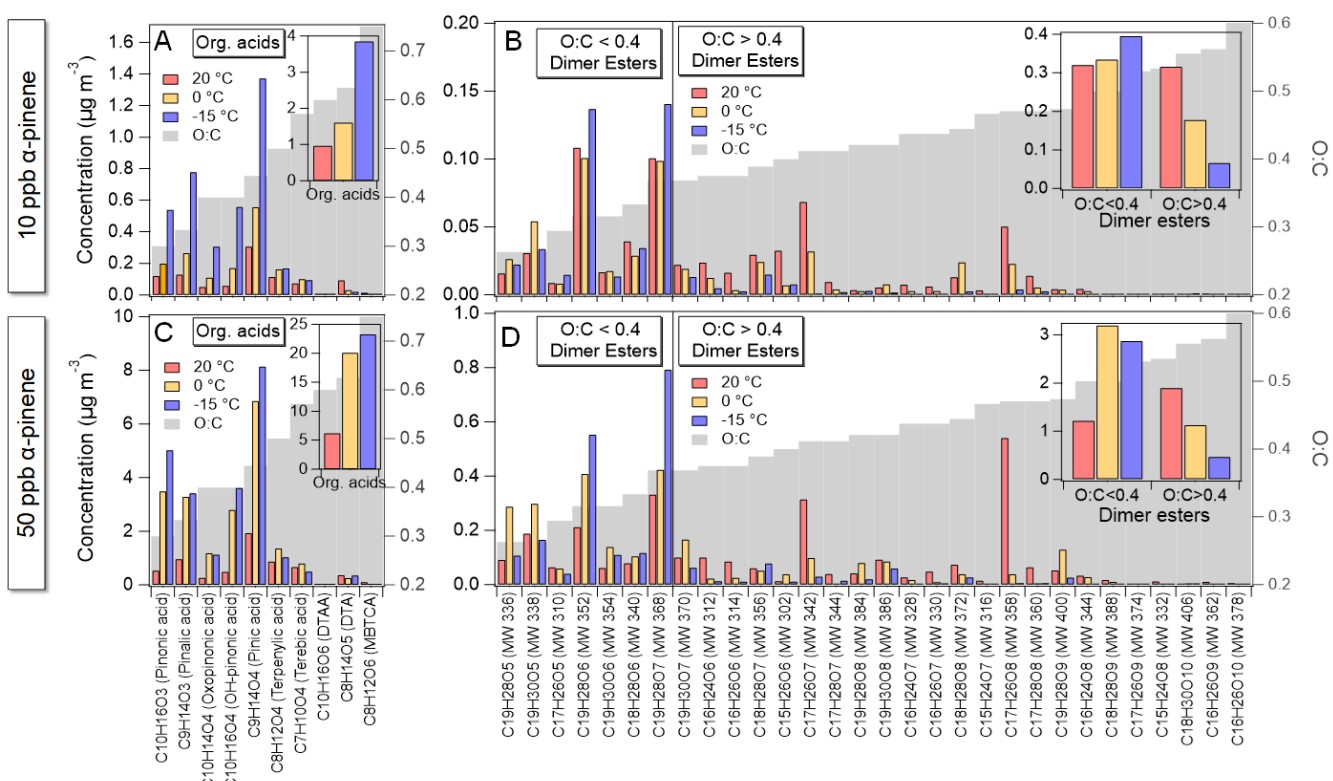

**Figure 5**. LC-MS results showing mass concentrations (µg m⁻³, left axis) of acids and dimer etsers as well as O:C ratios (grey bars, right axis) of these at 20 °C (red), 0 °C (orange), and -15 °C (blue) for the two α-pinene concentrations; 10 ppb (top panels **A** and **B**), 50 ppb (bottom panels **C** and **D**). Inserts show the total concentrations of the identified organic acids and dimer esters (O:C < 0.4 and O:C > 0.4)

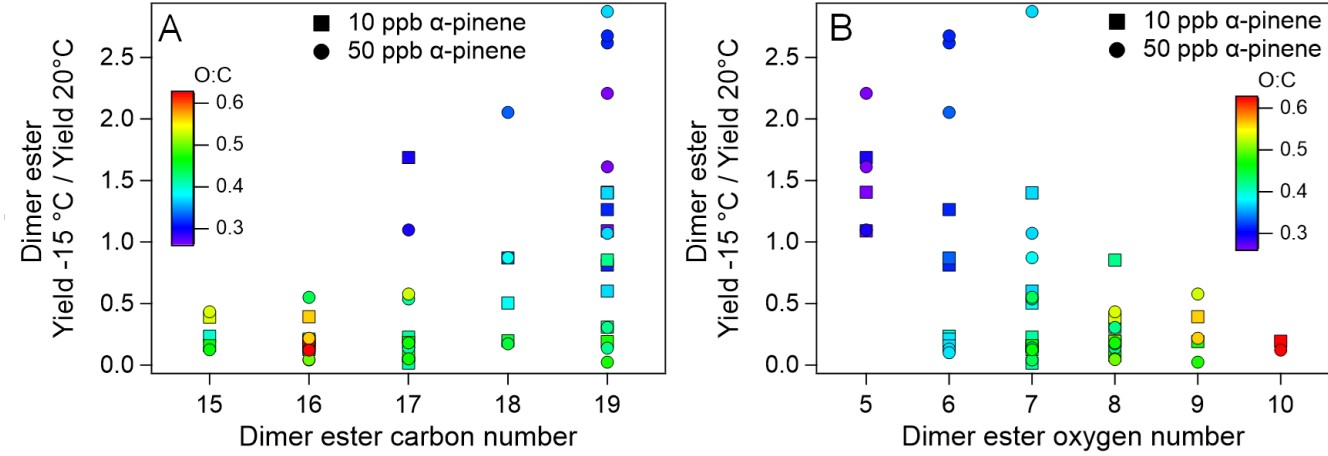

**Figure 6.** Comparison of relative yields (yield at -15 °C / yields at 20 °C) for specific dimer esters as a function of dimer ester carbon number (**A**) and oxygen number (**B**) in 10 and 50 ppb α-pinene ozonolysis experiments. Color scale indicates the O:C ratio of the dimer esters.

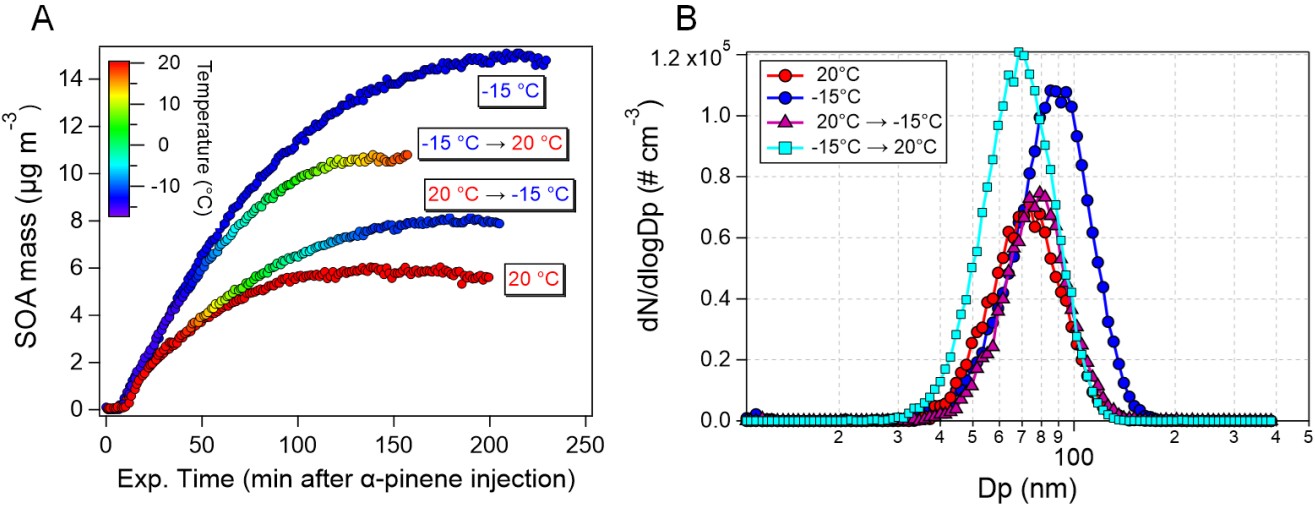

**Figure 7**. Effect of temperature ramping (20 °C → -15 °C, Exp. 1.4, and -15 °C → 20 °C, Exp. 1.5) on the wall-loss corrected SOA mass concentration (**A**) and final particle size distribution (**B**) from ozonolysis of 10 ppb α-pinene. For comparison, results from constant -15 °C and 20°C temperature experiments (Exp. 1.1 and Exp. 1.3, respectively) are also shown.

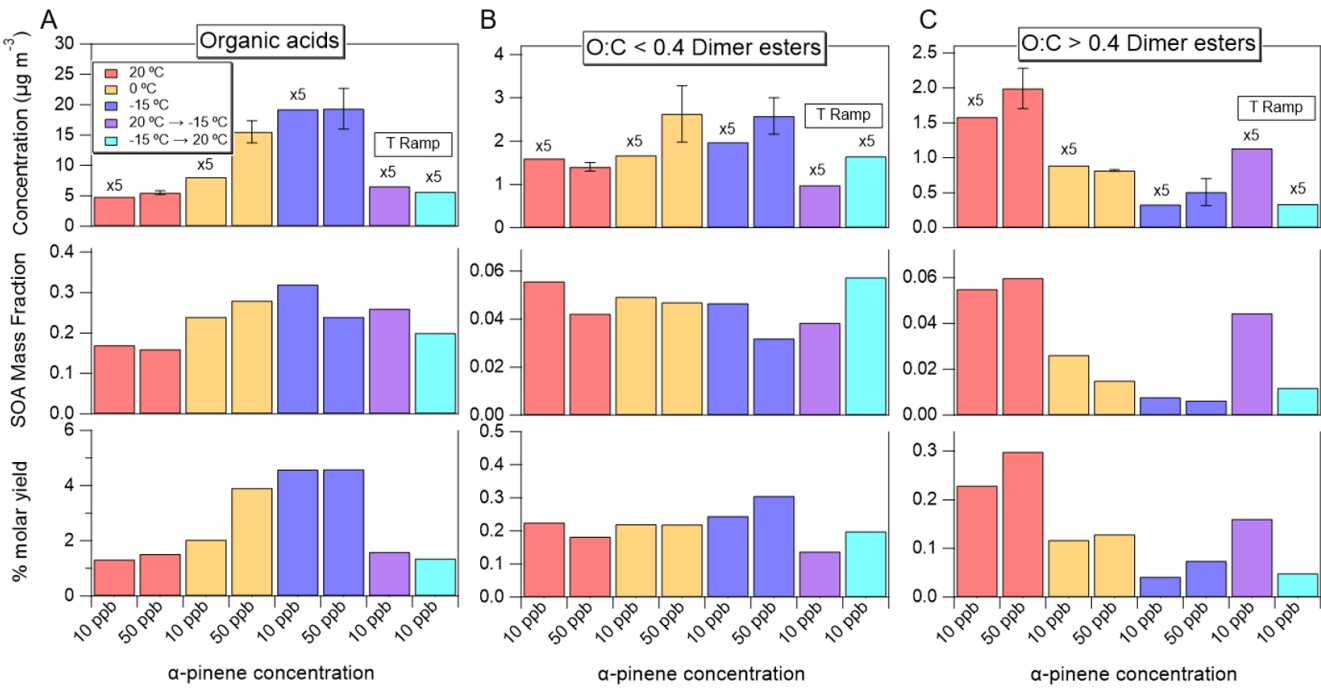

**Figure 8**. Effect of α-pinene concentration and temperature on the mass concentrations (μg m⁻³), SOA mass fractions, and molar yield of the identified organic acids (column A), low O:C (<0.4) dimer esters (column B) and high O:C (>0.4) dimer esters (column C). Note that the concentrations (μg m⁻³) of organic acids and dimer esters related to the 10 ppb α-pinene oxidation experiments have been multiplied with a factor of 5 (top panels). For the 50 ppb pinene oxidation experiments average values are reported from Exp. 2.1-2.3 and Exp. 3.1-3.3. Error bars represent one standard deviation.

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
