# Peer review of "The Aarhus Chamber Campaign on Highly Oxygenated Organic Molecules and Aerosols (ACCHA): Particle Formation, Organic acids, and Dimer Esters from Alpha-Pinene Ozonolysis at Different Temperatures"

_Atmospheric Chemistry and Physics, 2020_

## Referee Comment (RC1) · Anonymous Referee #1 · 7 Mar 2020

Kristensen et al. conducted dark ozonolysis of alpha pinene at different temperatures. They found at low temperatures the particle formation rates, particle number and mass concentrations were enhanced. The effects of temperature on particle-phase organic acids and dimer esters were also evaluated. They found the formation of less oxidized dimer esters increased at lower temperatures while the formation of the more oxidized species was suppressed. The link between the dimer ester and HOMs was also discussed. Though the topic of this study is timely and highly demanded in the area of

new particle formation, the manuscript is well written, and the results are interesting, I have some major concerns to be addressed before the recommendation of publication. The first major concern is the experiments were conducted at not dry conditions but the effects of RH on results were missing when the authors interpreted the results. The second major concern is the formation pathway of esters proposed by the authors is not fully convincing and other possible formation pathways found by other research groups are not mentioned.

Major comments: (1) Some statements in the Introduction section are not comprehensive and some related important references are missing, for example: Page 2, Line 50-51, Line 59-61: "higher yields with lower temperatures". This may not be absolutely right, as relative humidity, phase state, and the involved multiphase reactions all affect the SOA mass yields at different temperatures. For instance, von Hessberg et al. (2009) found the oscillatory negative temperature dependence under dry conditions while the positive temperature dependence under humid conditions when examining the SOA yield from the ozonolysis of beta pinene. Pathak et al. (2007) showed that the alpha pinene SOA yields showed a weak dependence on temperature in the 15°C to 40°C range, implying that the negative dependence of the partitioning on the temperature is counteracted by a positive dependence of the chemical mechanism (Tillmann et al., 2010). Page 2, Line 58: for organic compound partitioning, besides volatility and total particle mass, particle size and particle phase state also affect the partitioning and the gas-particle equilibration timescale significantly (Shiraiwa & Seinfeld, 2012; Liu et al., 2016; Li & Shiraiwa, 2019; Zaveri et al., 2014, 2020). Page 2, Line 72: Add Lawler et al. (2018). As far as I know, it is the first paper showing newly formed 20–70 nm particles showed enhancement in alkanoic acids from the perspective of ambient observations.

(2) Experiment section, Page 5, Line 187: As important results of this study are related to dimer esters and organic acids, what are the uncertainties in the estimations of the functional groups within SOA? The authors stated that due to lack of authentic

standards, the dimer esters were quantified using DTAA as surrogate standard. What is the effect of using DTAA other than another surrogate, e.g., bis(2-ethylhexyl) sebacate (Ranney & Ziemann, 2016; Claflin et al., 2018)?

(3) It is nice to show the various relative humidity values in Table 1 but the role of RH in the particle phase state, gas-particle partitioning, multiphase reactions (refer to the papers listed in Major Comment 1) and the concentrations of OH and HO2 radicals is not discussed in the manuscript, which could affect the interpretation of the experimental results. For example, Line 221, would the phase state also play a role (maybe a minor role though) making the maximum SOA mass reached slower at the lower temperature? Would the RH affect the detected concentrations of functional groups (Claflin et al., 2018) in Sec. 3.3? As the OH scavenger seems not used, would the RH affect the HO2 concentrations and thus the competition with the RO2-RO2 reaction (Claflin et al., 2018; Simon et al., 2020)?

(4) Figure 9: Is the reaction with RO2 radicals the only termination step? As far as I know, additional termination reactions of CI include their reactions with aldehydes (RC = O), alcohols (ROH), carboxylic acids (ROOH) to form hydroperoxy esters and secondary ozonides and water vapor to form hydroxy hydroperoxides (Claflin et al., 2018; Zeng et al., 2020). Claflin et al. (2018) stated that the only known gas-phase mechanism for forming esters in their experiments was the reaction of CI with carboxylic acids under dry conditions. Claflin et al. (2018) showed that particle-phase oxidation of carbonyl groups may contribute significantly to the formation of both carboxyl and ester groups in the SOA. Müller et al. (2008) have suggested that the esters they identified by mass spectrometry in SOA formed by $\alpha$-pinene ozonolysis might be formed by an unknown gas-phase radical mechanism. Could the authors give the reasons why other pathways can be excluded in Fig. 9? As the formation of esters via particle-phase decarboxylation of diacyl peroxides as proposed by Zhang et al. (2015) requires an ionic aerosol matrix such as aqueous ammonium sulfate, could the authors convince the readers the mechanisms proposed by Zhang et al. (2015) indeed happen in their

experiments without the presence of inorganic seeds?

Minor comments:

(1) P2, Line 55: Cite a review paper (e.g., Nizkorodov et al., 2011; Noziere et al., 2015) after "…peroxides and peroxy-acids". (2) P5, Line 163: Does the temperature gradually and continuously change or the system stay for some time at each temperature? In Figure S1, for the panel for Exp. 1.5, move "Temp" closer to the temperature line instead of the RH line. (3) Was OH scavenger used in the experiments? (4) Reference list: P19, Line 547: The title of Jensen et al. (2020) is Temperature and VOC concentration as controlling factors for chemical composition of alpha-pinene derived secondary organic aerosol: https://www.atmos-chem-phys-discuss.net/acp-2020-100/. P21, Line 613-614: Peräkylä et al. has been published in ACP.

References:

Claflin, M. S., Krechmer, J. E., Hu, W., Jimenez, J. L. and Ziemann, P. J.: Functional Group Composition of Secondary Organic Aerosol Formed from Ozonolysis of $\alpha$-Pinene Under High VOC and Autoxidation Conditions, ACS Earth and Space Chemistry, 2, 1196-1210, 10.1021/acsearthspacechem.8b00117, 2018.

Lawler, M. J., Rissanen, M. P., Ehn, M., Mauldin, R. L., Sarnela, N., Sipilä, M., & Smith, J. N. ( 2018). Evidence for diverse biogeochemical drivers of boreal forest new particle formation. Geophysical Research Letters, 45, 2038– 2046. https://doi.org/10.1002/2017GL076394

Li, Y. and Shiraiwa, M.: Timescales of secondary organic aerosols to reach equilibrium at various temperatures and relative humidities, Atmos. Chem. Phys., 19, 5959-5971, 10.5194/acp-19-5959-2019, 2019.

Liu, P., Li, Y. J., Wang, Y., Gilles, M. K., Zaveri, R. A., Bertram, A. K. and Martin, S. T.: Lability of secondary organic particulate matter, Proc. Natl. Acad. Sci. U.S.A., 113, 12643-12648, 10.1021/acscentsci.7b00452, 2016.

Mohr, C., Thornton, J. A., Heitto, A., Lopez-Hilfiker, F. D., Lutz, A., Riipinen, I., Hong, J., Donahue, N. M., Hallquist, M. and Petäjä, T.: Molecular identification of organic vapors driving atmospheric nanoparticle growth, Nature communications, 10, 1-7, 2019.

Müller, L., Reinnig, M.-C., Warnke, J., and Hoffmann, Th.: Unambiguous identification of esters as oligomers in secondary organic aerosol formed from cyclohexene and cyclohexene/$\alpha$-pinene ozonolysis, Atmos. Chem. Phys., 8, 1423–1433, https://doi.org/10.5194/acp-8-1423-2008, 2008.

Nizkorodov, S. A., Laskin, J. and Laskin, A.: Molecular chemistry of organic aerosols through the application of high resolution mass spectrometry, Physical Chemistry Chemical Physics, 13, 3612-3629, 10.1039/c0cp02032j, 2011.

Noziere, B., Kaberer, M., Claeys, M., Allan, J., D'Anna, B., Decesari, S., Finessi, E., Glasius, M., Grgic, I., Hamilton, J. F., Hoffmann, T., Iinuma, Y., Jaoui, M., Kahno, A., Kampf, C. J., Kourtchev, I., Maenhaut, W., Marsden, N., Saarikoski, S., Schnelle-Kreis, J., Surratt, J. D., Szidat, S., Szmigielski, R. and Wisthaler, A.: The Molecular Identification of Organic Compounds in the Atmosphere: State of the Art and Challenges, Chem. Rev., 115, 3919-3983, 10.1021/cr5003485, 2015.

Pathak, R. K., Stanier, C. O., Donahue, N. M., and Pandis, S. N. ( 2007), Ozonolysis of $\alpha$‐pinene at atmospherically relevant concentrations: Temperature dependence of aerosol mass fractions (yields), J. Geophys. Res., 112, D03201, doi:10.1029/2006JD007436.

April P. Ranney & Paul J. Ziemann (2016) Microscale spectrophotometric methods for quantification of functional groups in oxidized organic aerosol, Aerosol Science and Technology, 50:9, 881-892, DOI: 10.1080/02786826.2016.1201197

Shiraiwa, M. and Seinfeld, J. H.: Equilibration timescale of atmospheric secondary organic aerosol partitioning, Geophys. Res. Lett., 39, 10.1029/2012GL054008, 2012.

Simon, M., et al.: Molecular understanding of new-particle formation from alphapinene between $-50$ °C and $25$ °C, Atmos. Chem. Phys. Discuss., https://doi.org/10.5194/acp-2019-1058, 2020.

Tillmann, R., Hallquist, M., Jonsson, Å. M., Kiendler-Scharr, A., Saathoff, H., Iinuma, Y., and Mentel, Th. F.: Influence of relative humidity and temperature on the production of pinonaldehyde and OH radicals from the ozonolysis of $\alpha$-pinene, Atmos. Chem. Phys., 10, 7057–7072, https://doi.org/10.5194/acp-10-7057-2010, 2010.

von Hessberg, C., von Hessberg, P., Pöschl, U., Bilde, M., Nielsen, O. J. and Moortgat, G. K.: Temperature and humidity dependence of secondary organic aerosol yield from the ozonolysis of $\beta$-pinene, Atmos. Chem. Phys., 9, 3583-3599, 10.5194/acp-9-3583-2009, 2009.

Zaveri, R. A., Easter, R. C., Shilling, J. E. and Seinfeld, J. H.: Modeling kinetic partitioning of secondary organic aerosol and size distribution dynamics: representing effects of volatility, phase state, and particle-phase reaction, Atmos. Chem. Phys., 14, 5153-5181, 10.5194/acp-14-5153-2014, 2014.

Zaveri, R. A., Shilling, J. E., Zelenyuk, A., Zawadowicz, M. A., Suski, K., China, S., Bell, D. M., Veghte, D. and Laskin, A.: Particle-Phase Diffusion Modulates Partitioning of Semivolatile Organic Compounds to Aged Secondary Organic Aerosol, Environmental Science & Technology, 54, 2595-2605, 10.1021/acs.est.9b05514, 2020.

Zeng, M., Heine, N. and Wilson, K. R.: Evidence that Criegee intermediates drive autoxidation in unsaturated lipids, Proceedings of the National Academy of Sciences, 117, 4486-4490, 10.1073/pnas.1920765117, 2020.

Zhang, X., McVay, R. C., Huang, D. D., Dalleska, N. F., Aumont, B., Flagan, R. C. and Seinfeld, J. H.: Formation and evolution of molecular products in $\alpha$-pinene secondary organic aerosol, Proceedings of the National Academy of Sciences, 112, 14168-14173, 10.1073/pnas.1517742112, 2015.

---

## Referee Comment (RC2) · Anonymous Referee #2 · 26 Mar 2020

General comments: This work by Kristensen et al. studied ïĄą-pinene SOA formation and composition at different temperatures (20ïĆřC, 0ïĆřC, and -15ïĆřC) in a chamber facility. This study examined organic acids and dimer esters in the SOA composition through off-line LC-MS analysis. These chemicals were estimated to account for substantial fractions (15 – 30% and 4 – 11%, respectively) of total SOA mass. Dimers with lower O:C ratios (< 0.4) were found to increase at lower temperatures. In temperature ramping experiments, SOA mass and composition were found to be governed mostly

by initial temperatures. Overall, the manuscript is well written and demonstrates new findings regarding temperature effects on ïĄą-pinene SOA composition on the molecular level, especially at very low temperatures. But a few major concerns need to be addressed before this manuscript can be considered publishing.

Specific comments: 1. This manuscript has several companion papers published, as mentioned in the Introduction. If comparisons will be made with these papers, I suggest adding a section in the Results that briefly describes the main findings of these companion studies relevant to this work would be helpful.

2. Line 167 - 169. The influence of injection flow rate was not motivated clearly. Why did the authors think changing injection flow rate could affect the experiments? Without clear motivation, this part should be removed.

3. What is the scientific basis that made the authors to use O:C ratios of 0.4 as the threshold? What if one chooses 0.5? Instead of using an arbitrary value, showing histogram as a function of O:C ratios (by 0.1 increment) might be better.

4. Line 290 – 297. The authors observed different results compared to Kourtchev et al. (2016). The explanation should be explained to some extent. For example, whether the higher SOA mass loading under higher VOC lead to condensation of SVOC, which as a result lower oligomer fraction, as a competing process with the mechanism presented by Kourtchev et al. (2016).

5. As direct comparisons were made for the LCMS measured organic acids and dimer esters between different temperature conditions, one would expect that the quantified concentrations are reproducible and the relative abundance between different conditions are reproducible as well. In the current form of the manuscript, the reproducibility or uncertainty range was not discussed and should be addressed in the revised manuscript (e.g., error bars on Figures 5 and 6 representative of reproducibility).

6. Line 312 – 317. The authors argue that higher O:C dimers are formed through RO2-

RO2 reactions, followed by diacyl peroxide decomposition; while lower O:C dimers could be diacyl peroxides. However, it is not necessary that diacyl peroxides have lower O:C ratios than their decomposition products (loss of CO2). This argument needs to be better justified.

7. Section 3.4. After temperature ramping to 20 ïĆřC or -15ïĆřC, the SOA mass do not merge to the level in constant 20 ïĆřC or -15ïĆřC. The similar temperature effects have been studied in prior studies (Warren et al. 2009; Zhao et al., 2019). Zhao et al. (2019) provided some possible explanations for this behavior. The molecular results here, are likely better quantified and thus are in better position to explore more on the mechanistic explanation. However, it is missing from this section in the current form, except that the authors claimed the initial temperatures play a bigger role in final SOA mass.

8. From the title, it appears linking HOMs with organic acids and dimer esters is a key subject for this study. However, the manuscript discussed very little on this connection (only Section 3.5). The results of the referred companion study using NO3-CIMS should be discussed more extensively. Further, it is true that at lower temperature, HOM formation via RO2 autoxidation is limited, bimolecular RO2 reaction is expected to increase. But this does not necessarily mean that RO2-RO2 dimer formation is going to be enhanced. How about RO2 + HO2 and RO2 + RO2 which lead to monomeric products? These two reactions are both temperature dependent and are likely more important than RO2 autoxidation (5-10%) and RO2 + RO2 could more likely govern the changes in RO2 autoxidation and dimer formation. It is a four-factor relationship, but only the two less dominant pathways are discussed. In addition, as pointed by the authors, RO2 + RO2 might only explain some of the dimer esters.

Technical comments: 1. Line 41. Add "(SOA)" followed by "secondary organic aerosol". With this change, the "secondary organic aerosol" at Line 45 could be removed.

2. Line 43 - 46. ïĄą-pinene is also dominant OA source at other locations. For example, Zhang et al. 2018, 115, 2038, PNAS and Lee and Thornton et al., 2020, ACS Earth and Space Chem. (in press) show monoterpene SOA are the largest sources of PM in the southeastern US.

3. Line 49. A new study (Zhao et al., 2019, 3, 2549, ACS Earth and Space Chem.) performed similar temperature-ramping experiments with compositional analysis like this work and should be added in this list and perhaps later discussion (Section 3.4).

4. Line 130 - 146. Are the suite of online instrumentation situated in the cold room as well? It should be provided and if not, potential influence caused by temperature variation should be discussed.

5. Line 150 – 154. Two sentences have repeated texts. Please reword.

6. Line 161. This sentence should clarify if the temperature ramping started before or after SOA formation reached plateau.

7. Section 2.1. Slight RH variations between different temperature conditions are shown in Table 1, but should also be mentioned (one sentence) in the description.

---

## Author Comment (AC1) · 6 Jul 2020

Reply to review by Anonymous Referee #1

We thank the reviewer for the constructive comments, which we have addressed in a point-by-point fashion below. We have modified the manuscript accordingly.

Kristensen et al. conducted dark ozonolysis of alpha pinene at different temperatures.

[Figure]

They found at low temperatures the particle formation rates, particle number and mass concentrations were enhanced. The effects of temperature on particle-phase organic acids and dimer esters were also evaluated. They found the formation of less oxidized dimer esters increased at lower temperatures while the formation of the more oxidized species was suppressed. The link between the dimer ester and HOMs was also discussed. Though the topic of this study is timely and highly demanded in the area of new particle formation, the manuscript is well written, and the results are interesting, I have some major concerns to be addressed before the recommendation of publication. The first major concern is the experiments were conducted at not dry conditions but the effects of RH on results were missing when the authors interpreted the results. The second major concern is the formation pathway of esters proposed by the authors is not fully convincing and other possible formation pathways found by other research groups are not mentioned.

Major comments: (1) Some statements in the Introduction section are not comprehensive and some related important references are missing, for example: Page 2, Line 50-51, Line 59-61: "higher yields with lower temperatures". This may not be absolutely right, as relative humidity, phase state, and the involved multiphase reactions all affect the SOA mass yields at different temperatures. For instance, von Hessberg et al. (2009) found the oscillatory negative temperature dependence under dry conditions while the positive temperature dependence under humid conditions when examining the SOA yield from the ozonolysis of beta pinene. Pathak et al. (2007) showed that the alpha pinene SOA yields showed a weak dependence on temperature in the 15◦C to 40◦C range, implying that the negative dependence of the partitioning on the temperature is counteracted by a positive dependence of the chemical mechanism (Tillmann et al., 2010). Page 2, Line 58: for organic compound partitioning, besides volatility and total particle mass, particle size and particle phase state also affect the partitioning and the gas-Pa$\frac{1}{2}$rticle equilibration timescale significantly (Shiraiwa&Seinfeld,2012; Liu et al., 2016; Li & Shiraiwa, 2019; Zaveri et al., 2014, 2020). Page 2, Line 72: Add Lawler et al. (2018). As far as I know, it is the first paper showing newly formed 20–70 nm particles showed enhancement in alkanoic acids from the perspective of ambient observations.

Reply: We are grateful for the additional information and references provided by the reviewer and we have added this to the manuscript in the following sentences:

Line 64-66: "The extent and timescale to which an organic compound undergoes partitioning is related to its saturation vapor pressure, the available particle mass (Kroll and Seinfeld, 2008) as well as particle size and particle phase (Shiraiwa and Seinfeld, 2012;Li and Shiraiwa, 2019;Zaveri et al., 2014;Zaveri et al., 2020)."

Line 73-79: "In addition to temperature-dependent condensation of oxidation products, temperature-modulated gas and multiphase chemistry has been suggested to influence SOA yields from VOC oxidation. Von Hessberg et al. (2009) observed oscillatory positive temperature dependence under dry conditions and suggested that SOA yields from $\beta$-pinene oxidation is governed to a higher degree by the temperature and humidity dependence of the involved chemical reactions than by vapor pressure of the formed oxidation products at different temperatures. Furthermore, Pathak et al. (2007a) observed that $\alpha$-pinene SOA yields showed a weak dependence on temperature in the 15 °C to 40 °C range, implying that the negative temperature dependence of the partitioning is counteracted by a positive dependence of the chemical reaction mechanism."

Line 86-87: "In relation, Lawler et al. (2018) observed enhanced content of alkanoic acids in newly formed 20–70 nm particles in the Finnish boreal forest."

(2) Experiment section, Page 5, Line 187: As important results of this study are related to dimer esters and organic acids, what are the uncertainties in the estimations of the functional groups within SOA? The authors stated that due to lack of authentic standards,the dimer esters were quantified using DTAA as surrogate standard. What is the effect of using DTAA other than another surrogate, e.g., bis(2-ethylhexyl) sebacate (Ranney & Ziemann, 2016; Claflin et al., 2018)?

Reply: As the majority of the organic acids are identified and quantified using authentic standards, the uncertainties related to the reported data is expected to be low (< 20 %). With respect to dimer ester, a surrogate (DTAA) is used which shows good resemblance to the dimer esters both in chemical structure and UHPLC retention time (as the signal response is affected by eluent composition).

The following sentences has been added to the manuscript (Line 204-206): "The analytical uncertainty is estimated to be < 20 % for carboxylic acids. Due to lack of authentic standards, the dimer esters were quantified using DTAA as surrogate standard. DTAA was chosen due to its structural similarities with that of the dimer esters (dicarboxylic acid with ester functionality) as well as similar UHPLC retention time."

It is difficult to estimate the uncertainties associated with the use of surrogates without available authentic standards for comparison. Using bis(2-ethylhexyl) sebacate is not advisable when quantifying the dimer esters reported in the current manuscript as this compound is structurally and functionally very different from the dimer esters (i.e. contains no acid functionalities) and would thus result in a very different response in the UHPLC-MS system adding to significant uncertainties. With respect to the overall uncertainties related to reproducibility of the performed UHPLC-MS analysis we have added a new figure in SI comparing the UHPLC-MS results of experiments performed at similar conditions. The comparison show < 10 % variation between results from similar experiments. This has been added to the manuscript:

Line 283-284: "From comparison of repeated experiments (Exp. 2.1 - 2.3 and Exp. 3.1 - 3.3) the uncertainties related to the presented UHPLC/ESI-qTOF-MS results are estimated to be less than 10 %, Figure S3)."

(3) It is nice to show the various relative humidity values in Table 1 but the role of RH in the particle phase state, gas-particle partitioning, multiphase reactions (refer to the papers listed in Major Comment 1) and the concentrations of OH and HO2 radicals is not discussed in the manuscript, which could affect the interpretation of

the experimental results. For example, Line 221, would the phase state also play a role (maybe a minor role though) making the maximum SOA mass reached slower at the lower temperature? Would the RH affect the detected concentrations of functional groups (Claflin et al., 2018) in Sec. 3.3? As the OH scavenger seems not used, would the RH affect the HO2 concentrations and thus the competition with the RO2-RO2 reaction (Claflin et al., 2018; Simon et al., 2020)?

Reply: It is true that RH could affect the growth of SOA particles through changes of the particle viscosity as reported in Zaveri et al (2020) and consequently result in RH-modulated portioning of SVOCs to the particle phase. However, as shown in Table 1, the RH values reported for all experiments are relatively low (<20 %), even at -15 °C, and so are the differences in RH between experiments, thus we do not believe that these differences would have a significant effect. This is supported by Zaveri et al. (2020) stating that "The diffusivity within the aged $\alpha$-pinene SOA remains appreciably slow even at 80% RH". However, we have added the following discussion on the particle phase on condensation and evaporation of SVOCs to explain the results from temperature ramping experiments:

Line 370-382 "In contrast, cooling of the formed SOA particles in Exp.1.4 resulted only in a small 0.3 $\mu$g m-3 increase in organic acid concentration making up $\sim$ 15 % of the reported 2 $\mu$g m-3 increase in SOA mass compared to the constant 20 °C experiment. These results indicates limited gas-to-particle phase condensation of the semi-volatile organic acids upon cooling the SOA. The current findings are in agreement with that of Zhao et al. (2019) who found that during cooling of $\alpha$-pinene SOA particles the resulting mass growth was largely over predicted by the applied volatility basis set model. As suggested in Zhao et al. (2019) the observed limited condensation of semi-volatiles to SOA particles during cooling is likely attributed to an expected high particle viscosity at the relative low RH conditions of the performed oxidation experiments. Under these conditions, condensed SVOCs are likely confined at the near-particle-surface region thus impeding further partitioning of the gas-phase SVOCs (Renbaum-

Wolff et al., 2013). In the current study, this is supported by the relatively small increase in organic acids (Fig. 8) as well as almost negligible increase in particle size (Fig. 7B) observed upon cooling the SOA particles from 20 °C to -15 °C. Also, SVOCs reside near the particle surface, this would indeed explain the effective evaporation of these compounds (i.e. the identified organic acids, Fig. 8) and subsequent reduction in particle size (Fig. 7B) observed in Exp. 1.5."

Furthermore, we have expanded section 3.5 in which the dimer ester formation is discussed to include formation pathways suggested by other research groups including reactions of stabilized Criegee intermediates with oxidation products and RO-RO2 reactions. As suggested by the reviewer the influence of HO2 + RO2 competition is also discussed:

Line 459-469:"The RO2–RO2 reaction is expected to compete with the reactions of RO2 with HO2 radicals. In the absence of an OH-scavenger, the performed oxidation experiments will include the formation of OH-radicals from the gas-phase reaction of $\alpha$-pinene with O3. The formed OH-radicals reacts readily with O3 yielding HO2-radicals available for RO2-HO2 reactions. As both reactions (O3 + $\alpha$-pinene and O3 + OH) have a positive temperature dependence, the formation of HO2 and its subsequent reaction with RO2 is expected to increase at the higher reaction temperatures. In Simon et al. (2020) the higher concentration of HO2 leads to an increased competition with the RO2-RO2 self-reaction, which reduced the formation of HOM dimers but increased HOM monomers. In the current study, however, reduction in the concentration of dimer esters due to increased RO2-HO2 competition at higher temperatures is only observed in the case of the low O:C dimer esters. In the case of the higher O:C dimers, it appears that a suppressed competition of HO2 with RO2 at the lower temperatures is less important compared to the reduced availability of more oxidized species for dimer ester formation. "

(4) Figure 9: Is the reaction with RO2 radicals the only termination step? As far as I know, additional termination reactions of CI include their reactions with aldehydes

(RC = O), alcohols (ROH), carboxylic acids (ROOH) to form hydroperoxy esters and secondary ozonides and water vapor to form hydroxy hydroperoxides (Claflin et al., 2018; Zeng et al., 2020). Claflin et al. (2018) stated that the only known gas-phase mechanism for forming esters in their experiments was the reaction of CI with carboxylic acids under dry conditions. Claflin et al. (2018) showed that particle-phase oxidation of carbonyl groups may contribute significantly to the formation of both carboxyl and ester groups in the SOA. Müller et al. (2008) have suggested that the esters they identified by mass spectrometry in SOA formed by $\alpha$-pinene ozonolysis might be formed by an unknown gas-phase radical mechanism. Could the authors give the reasons why other pathways can be excluded in Fig. 9? As the formation of esters via particle-phase decarboxylation of diacyl peroxides as proposed by Zhang et al. (2015) requires an ionic aerosol matrix such as aqueous ammonium sulfate, could the authors convince the readers the mechanisms proposed by Zhang et al. (2015) indeed happen in their experiments without the presence of inorganic seeds?

Reply: We agree with the reviewer that the mechanism proposed by Zhang et al. (2015) is unlikely in the absence of an ionic aerosol matrix. Thus, we have rewritten the discussion of dimer ester formation to include multiple formation pathways suggested by other research groups including reactions of CI with oxidation products with various functionalities. As we cannot exclude other formation pathways (i.e. sCI reactions) Figure 9 has been revised to the possibilities of different reaction pathways. Section 3.5 (revised): "Gas-phase formation of dimer esters from $\alpha$-
[revised manuscript text omitted]
 $\alpha$-pinene with O3. The formed OH-radicals reacts readily with O3 yielding HO2-radicals available for RO2-HO2 reactions. As both reactions (O3 + $\alpha$-pinene and O3 + OH) have a positive temperature dependence, the formation of HO2 and its subsequent reaction with RO2 is expected to increase at the higher reaction temperatures. In Simon et al. (2020) the higher concentration of HO2 leads to an increased competition with the RO2-RO2 self-reaction, which reduced the formation of HOM dimers but increased HOM monomers. However, in the current study, reduction in the concentration of dimer esters due to increased RO2-HO2 competition at higher temperatures is only observed in the case of the low O:C dimer esters. In the case of the higher O:C dimers, it appears that a suppressed competition of HO2 with RO2 at the lower temperatures is less important compared to the reduced availability of more oxidized species for dimer ester formation. We propose that, although different in chemical structures and O:C-ratios, dimer esters and HOMs may be linked via their formation mechanisms, both involving RO2 autoxidation. The particle-phase dimer esters and the gas-phase HOMs may merely represent two different fates of the RO2 radicals. If conditions are favorable and efficient autoxidation takes place, this will result in the formation of HOMs, which by the definition recommended by Bianchi et al. (2019) in this case means any molecule with 6 or more oxygen atoms that has undergone autoxidation. On the other hand, dimer esters could be the product of RO2 cross reactions or reactions of sCI with the autoxidation termination products with O:C ratios influenced by the number of potential autoxidation steps undertaken by the involved RO2 species prior to reaction or termination (Fig. 9). Whether the formation of dimer esters proceeds through ROOR dimer formation from RO2-RO2 cross reactions or through monomeric compounds reacting with sCI is yet to be determined. Lastly, thermodynamics need to be considered as a possible explanation for the observed temperature responses of the high and low O:C dimer esters. As reported in Kristensen et al. (2017), the identified dimer esters span across a wide range of volatilities. Here, many of the low O:C dimer esters may be sufficiently volatile to allow

considerable fractions to exist in the gas phase at high temperature. Consequently, the increased particle-phase concentration observed at the lower temperatures may solely be attributed to enhanced gas-to-particle phase partitioning of these species. Supporting this, Mohr et al. (2017) identified dimeric monoterpene oxidation products (C16–20HyO6–9) in both particle and gas phases in ambient air measurements in the boreal forest in Finland."

Minor comments: (1) P2, Line 55: Cite a review paper (e.g., Nizkorodov et al., 2011; Noziere et al., 2015) after "...peroxides and peroxy-acids".

Reply: The suggested papers are now cited as suggested.

(2) P5, Line 163: Does the temperature gradually and continuously change or the system stay for some time at each temperature? In Figure S1, for the panel for Exp. 1.5, move "Temp" closer to the temperature line instead of the RH line.

Reply: The temperature changes gradually and continuously as evident from Figure S1 (Exp. 1.4 and 1.5). This is now stated in the sentence (Line 178-180): "In both experiments the gradual and continuous temperature ramping was initiated approximately 40 min after the injection of $\alpha$-pinene, hence before the SOA formation plateaued." Reply: Figure S1 (Exp. 1.5) has been changed as suggested.

(3) Was OH scavenger used in the experiments?

Reply: No. This is now clarified in the manuscript. The following sentence has been added (Line 186): "All experiments were performed without the addition of an OH-scavenger to the chamber."

(4) Reference list: P19, Line 547: The title of Jensen et al. (2020) is Temperature and VOC concentration as controlling factors for chemical composition of alpha-pinene derived secondary organic aerosol: https://www.atmos-chem-phys-discuss.net/acp-2020-100/. P21, Line 613-614: Peräkylä et al. has been published in ACP.

Reply: This has been corrected.

[Figure]

**Figure S3**. Concentrations ($\mu g\ m^{-3}$) of acids and dimers from UHPLC/ESI-qTOF-MS analysis of repeated experiments performed at 50 ppb α-pinene and 20 °C (Exp. 2.1 & 3.1), 0 °C (Exp. 2.2 & 3.2) and -15 °C (Exp. 2.3a, 2.3b & 3.3). Bars to the right (dark colored) represent average concentrations and associated standard deviations.

**Fig. 1.** Figure S3 added to SI

---

## Author Comment (AC2) · 6 Jul 2020

Reply to review by Anonymous Referee #2

We thank the reviewer for the constructive comments, which we have addressed in a point-by-point fashion below. We have modified the manuscript accordingly.

General comments: This work by Kristensen et al. studied ïAËŻaËŻ-pinene SOA for-

mation and composition at different temperatures (20℃, 0℃, and -15℃) in a chamber facility. This study examined organic acids and dimer esters in the SOA composition through off-line LC-MS analysis. These chemicals were estimated to account for substantial fractions (15 – 30% and 4 – 11%, respectively) of total SOA mass. Dimers with lower O:C ratios (< 0.4) were found to increase at lower temperatures. In temperature ramping experiments, SOA mass and composition were found to be governed mostly by initial temperatures. Overall, the manuscript is well written and demonstrates new findings regarding temperature effects on α-pinene SOA composition on the molecular level, especially at very low temperatures. But a few major concerns need to be addressed before this manuscript can be considered publishing.

Specific comments:

1. This manuscript has several companion papers published, as mentioned in the Introduction. If comparisons will be made with these papers, I suggest adding a section in the Results that briefly describes the main findings of these companion studies relevant to this work would be helpful.

Reply: As the comparison with companion papers are very limited in the manuscript confining only to a short reference to the findings by Quéléver et al. (2019) in section 3.5 we do not believe that further descriptions of the findings of the companion papers are needed – especially considering the length of the manuscript in its current form.

2. Line 167 – 169. The influence of injection flow rate was not motivated clearly. Why did the authors think changing injection flow rate could affect the experiments? Without clear motivation, this part should be removed.

Reply: This has been removed as suggested by the reviewer.

3. What is the scientific basis that made the authors to use O:C ratios of 0.4 as the threshold? What if one chooses 0.5? Instead of using an arbitrary value, showing

histogram as a function of O:C ratios (by 0.1 increment) might be better.

Reply: The value of 0.4 is derived from Figure 6 and is the O:C value above which all dimer esters show a decrease in concentration at -15°C compared to 20 °C. Unfortunately, this is not clearly stated in the manuscript. We have thus added a figure to the SI (Figure S4) showing dimer ester yields at -15°C relative to 20 °C (yield @ -15°C / yield @ 20°C) as a function of O:C-ratios and added a line showing how the 0.4 value (actually 0.38) is derived.

The following sentence has been added (Line 333-335): "Accordingly, the dimer esters are grouped based on their O:C ratios, with the more oxidized dimer esters having an O:C > 0.4; the O:C value above which all dimer esters show decreased concentration at the lower -15 °C compared to 20 °C (Figure S4)."

4. Line 290 – 297. The authors observed different results compared to Kourtchev et al. (2016). The explanation should be explained to some extent. For example, whether the higher SOA mass loading under higher VOC lead to condensation of SVOC, which as a result lower oligomer fraction, as a competing process with the mechanism presented by Kourtchev et al. (2016).

Reply: The comparison with Kourtchev et al. (2016) has been removed as the finding by Kourtchev et al. (2016) relates to the number of oligomeric compounds and not their concentrations, thus a direct comparison is not valid. However, we have added the following relating to Kourtchev et al. (2016):

Line 320-324: "In relation, Kourtchev et al. (2016) observed a positive relationship between temperature and oligomer fraction in aerosol samples collected at Hyytiälä in summer 2011 and 2014 but ascribed this to differences in the VOC emissions. However, the current study, indicates that temperature alone may influence the formation of dimeric compounds thus supporting to the ambient observation in Kourtchev et al. (2016)."

5. As direct comparisons were made for the LCMS measured organic acids and dimer esters between different temperature conditions, one would expect that the quantified concentrations are reproducible and the relative abundance between different conditions are reproducible as well. In the current form of the manuscript, the reproducibility or uncertainty range was not discussed and should be addressed in the revised manuscript (e.g., error bars on Figures 5 and 6 representative of reproducibility).

Reply: We agree with the reviewer in that the evidence of the reproducibility of the performed LCMS analysis is lacking in the current form of the manuscript. To remedy this, we have added a new figure in the SI comparing the LCMS results of experiments performed at similar conditions. The comparison shows < 10 % variation between LCMS results from similar experiments. This has been added to the manuscript:

Line 283-284: "From comparison of repeated experiments (Exp. 2.1 - 2.3 and Exp. 3.1 - 3.3) the uncertainties related to the presented UHPLC/ESI-qTOF-MS results are estimated to be less than 10 %, Figure S3)."

6. Line 312 – 317. The authors argue that higher O:C dimers are formed through RO2-RO2 reactions, followed by diacyl peroxide decomposition; while lower O:C dimers could be diacyl peroxides. However, it is not necessary that diacyl peroxides have lower O:C ratios than their decomposition products (loss of CO2). This argument needs to be better justified.

Reply: We agree with the reviewer and have removed the argument in question.

7. Section 3.4. After temperature ramping to 20 ïʹ CËĞrC or -15ïʹ CËĞrC, the SOA mass do not merge to the level in constant 20 ïʹ CËĞrC or -15ïʹ CËĞrC. The similar temperature effects have been studied in prior studies (Warren et al. 2009; Zhao et al., 2019). Zhao et al. (2019) provided some possible explanations for this behavior. The molecular results here, are likely better quantified and thus are in better position to explore more on the mechanistic explanation. However, it is missing from this section in the current form, except that the authors claimed the initial temperatures play a bigger

role in final SOA mass.

Reply: We agree with the reviewer and have added a discussion on the results from temperature ramping:

Line 370-382 "In contrast, cooling of the formed SOA particles in Exp.1.4 resulted only in a small 0.3 $\mu$g m-3 increase in organic acid concentration making up $\sim$ 15 % of the reported 2 $\mu$g m-3 increase in SOA mass compared to the constant 20 °C experiment. These results indicates limited gas-to-particle phase condensation of the semi-volatile organic acids upon cooling the SOA. The current findings are in agreement with that of Zhao et al. (2019) who found that during cooling of $\alpha$-pinene SOA particles the resulting mass growth was largely over predicted by the applied volatility basis set model. As suggested in Zhao et al. (2019) the observed limited condensation of semi-volatiles to SOA particles during cooling is likely attributed to an expected high particle viscosity at the relative low RH conditions of the performed oxidation experiments. Under these conditions, condensed SVOCs are likely confined at the near-particle-surface region thus impeding further partitioning of the gas-phase SVOCs (Renbaum-Wolff et al., 2013). In the current study, this is supported by the relatively small increase in organic acids (Fig. 8) as well as almost negligible increase in particle size (Fig. 7B) observed upon cooling the SOA particles from 20 °C to -15 °C. Also, SVOCs reside near the particle surface, this would indeed explain the effective evaporation of these compounds (i.e. the identified organic acids, Fig. 8) and subsequent reduction in particle size (Fig. 7B) observed in Exp. 1.5."

8. From the title, it appears linking HOMs with organic acids and dimer esters is a key subject for this study. However, the manuscript discussed very little on this connection (only Section 3.5). The results of the referred companion study using NO3-CIMS should be discussed more extensively. Further, it is true that at lower temperature, HOM formation via RO2 autoxidation is limited, bimolecular RO2 reaction is expected to increase. But this does not necessarily mean that RO2-RO2 dimer formation is going to be enhanced. How about RO2 + HO2 and RO2 + RO2 which lead to monomeric

products? These two reactions are both temperature dependent and are likely more important than RO2 autoxidation (5-10%) and RO2+RO2 to ROOR (5 – 10%). Thus, the temperature effects on RO2 + HO2 and RO2 + RO2 could more likely govern the changes in RO2 autoxidation and dimer formation. It is a four-factor relationship, but only the two less dominant pathways are discussed. In addition, as pointed by the authors, RO2 + RO2 might only explain some of the dimer esters.

Reply: We agree with the reviewer that the title may be misleading as the discussion on HOMs in relation to organic acids and dimers is relatively brief in the manuscript. We have thus changed the manuscript title to capture the content in a more accurate manner: "The Aarhus Chamber Campaign on Highly Oxidized Multifunctional Organic Molecules and Aerosols (ACCHA): Particle Formation, Organic acids, and Dimer Esters from Alpha-Pinene Ozonolysis at Different Temperatures

In relation, we have expanded the section (now section 3.5, see below) in which the dimer ester formation is discussed to include formation pathways suggested by other research groups including reactions of stabilized Criegee intermediates with oxidation products and RO-RO2 reactions. Also, as suggested by the reviewer the influence of HO2 + RO2 competition is also discussed.

[revised manuscript text omitted]

O3. The formed OH-radicals reacts readily with O3 yielding HO2-radicals available for RO2-HO2 reactions. As both reactions (O3 + $\alpha$-pinene and O3 + OH) have a positive temperature dependence, the formation of HO2 and its subsequent reaction with RO2 is expected to increase at the higher reaction temperatures. In Simon et al. (2020) the higher concentration of HO2 leads to an increased competition with the RO2-RO2 self-reaction, which reduced the formation of HOM dimers but increased HOM monomers. However, in the current study, reduction in the concentration of dimer esters due to increased RO2-HO2 competition at higher temperatures is only observed in the case of the low O:C dimer esters. In the case of the higher O:C dimers, it appears that a suppressed competition of HO2 with RO2 at the lower temperatures is less important compared to the reduced availability of more oxidized species for dimer ester formation. We propose that, although different in chemical structures and O:C-ratios, dimer esters and HOMs may be linked via their formation mechanisms, both involving RO2 autoxidation. The particle-phase dimer esters and the gas-phase HOMs may merely represent two different fates of the RO2 radicals. If conditions are favorable and efficient autoxidation takes place, this will result in the formation of HOMs, which by the definition recommended by Bianchi et al. (2019) in this case means any molecule with 6 or more oxygen atoms that has undergone autoxidation. On the other hand, dimer esters could be the product of RO2 cross reactions or reactions of sCI with the autoxidation termination products with O:C ratios influenced by the number of potential autoxidation steps undertaken by the involved RO2 species prior to reaction or termination (Fig. 9). Whether the formation of dimer esters proceeds through ROOR dimer formation from RO2-RO2 cross reactions or through monomeric compounds reacting with sCI is yet to be determined. Lastly, thermodynamics need to be considered as a possible explanation for the observed temperature responses of the high and low O:C dimer esters. As reported in Kristensen et al. (2017), the identified dimer esters span across a wide range of volatilities. Here, many of the low O:C dimer esters may be sufficiently volatile to allow considerable fractions to exist in the gas phase at high temperature. Consequently, the increased particle-phase concentration observed

at the lower temperatures may solely be attributed to enhanced gas-to-particle phase partitioning of these species. Supporting this, Mohr et al. (2017) identified dimeric monoterpene oxidation products (C16–20HyO6–9) in both particle and gas phases in ambient air measurements in the boreal forest in Finland."

Technical comments:

1. Line41. Add "(SOA)"followed by "secondary organic aerosol". With this change, the "secondary organic aerosol" at Line 45 could be removed.

Reply: This has been changed as suggested by the reviewer

2. Line 43 – 46. ïÄŻaË̇Ż-pinene is also dominant OA source at other locations. For example, Zhang et al. 2018, 115, 2038, PNAS and Lee and Thornton et al., 2020, ACS Earth and Space Chem. (in press) show monoterpene SOA are the largest sources of PM in the southeastern US.

Reply: The following sentence has been added (line 49-50):"In addition, (Lee et al., 2020;Zhang et al., 2018) show that monoterpene SOA are the largest sources of particulate matter in the southeastern US."

3. Line 49. A new study (Zhao et al., 2019, 3, 2549, ACS Earth and Space Chem.) performed similar temperature-ramping experiments with compositional analysis like this work and should be added in this list and perhaps later discussion (Section 3.4).

Reply: A reference to the work by Zhao et al., 2019 has been added to the list.

4. Line 130 – 146. Are the suite of online instrumentation situated in the cold room as well? It should be provided and if not, potential influence caused by temperature variation should be discussed.

Reply: The following sentences has been added Line 143-144: "Additional instrumentations are situated in air-conditioned (constant 20 °C) laboratory directly outside the cold room." Line 158-160: "SMPS and PSM measurements were performed as close

as possible to the cold room trough insulated tubing extending $\sim$ 40 and $\sim$ 10 cm from the cold room, respectively, thus minimizing residence time and potential influences caused by temperature variations."

5. Line 150 – 154. Two sentences have repeated texts. Please reword.

Reply: This has been corrected

6. Line 161. This sentence should clarify if the temperature ramping started before or after SOA formation reached plateau.

Reply: The sentence now reads (Line 178-180) : "In both experiments the temperature ramping was initiated approximately 40 min after the injection of $\alpha$-pinene, hence before the SOA mass formation plateaued"

7. Section 2.1. Slight RH variations between different temperature conditions are shown in Table 1, but should also be mentioned (one sentence) in the description.

Reply: The following sentence has been added (Line 181-183) "Note that small variations in RH (< 25 %) are observed in between all conducted experiments arising from heating or cooling of the dry chamber air (Table 1)."
* * *
[Figure]

Figure S3. Concentrations (µg m$^{-3}$) of acids and dimers from UHPLC/ESI-qTOF-MS analysis of repeated experiments performed at 50 ppb α-pinene and 20 °C (Exp. 2.1 & 3.1), 0 °C (Exp. 2.2 & 3.2) and -15 °C (Exp. 2.3a, 2.3b & 3.3). Bars to the right (dark colored) represent average concentrations and associated standard deviations.

**Fig. 1.** Figure S3 added to SI

[Figure]

**Figure S4**. Comparison of relative yields (yield at -15 °C / yields at 20 °C) for specific dimer esters as a function of dimer ester O:C ratio in 10 and 50 ppb α-pinene ozonolysis experiments. Vertical line indicates the O:C value (0.38) above which all dimer esters show a decrease in concentration at – 15 °C compared to 20 °C.

**Fig. 2.** Figure S4 added to SI

[Figure]

**Figure 9**. Illustration of two suggested mechanism for dimer esters: 1) RO$_2$-RO$_2$ radical reactions forming ROOR dimers (upper) or 2) reactions of stabilized Criegee Intermediates (sCI) with monomers from RO$_2$ termination reactions with RO$_2$ or HO$_2$ (lower). The O:C of the resulting dimer ester is governed by the degree of autoxidation before radical termination.

**Fig. 3.** New Figure 9

---

## Author Response (AR2)

**Reply to review by Anonymous Referee #2**

*We thank the reviewer for the constructive and insightful comments. We have have addressed the review comments in a point-by-point fashion below and modified the manuscript accordingly.*

The authors have well addressed most of my previous suggestions and the revised manuscript has been improved in general. I have only one addition comment: I feel that the discussion of the linkage between autoxidation and dimer ester formation is still largely based on speculations and the related conclusions should be very carefully re-worded. Here are some points not fully explained by the authors' hypothesis:

*Reply: We agree with the reviewer that based on the presented results no clear conclusion on autoxidation and dimer formation can be made. The studied system is highly complex and interconnected and further kinetic studies designed to more accurately examine dimer ester formation is needed before ruling out or confirming any specific formation mechanism. For this reason, we do not strongly claim either one direction or the other and have thus removed Figure 9 from the manuscript as it insufficiently illustrated the multiple formation mechanisms that could possibly lead to dimer ester formation.*

1. As VOC concentration increased by 5 times, the RO2 + RO2 pathway should outcompete RO2 autoxidation much more significantly under all temperatures. This is not evident by the dimer ester comparisons from both the O:C<0.4 and O:C>0.4 groups. This observation in fact suggests that many of the dimer esters have nothing to do with RO2 autoxidation (or the competition processes). It is perhaps worthwhile identifying which dimer esters are least scaled with the VOC enhancement and those might be possibly related to autoxidation. The way that dimer esters are grouped here might be misleading.

*Reply: This is a great comment and observation by the reviewer. However, it is worth noting that that when alpha-pinene increases by 5x, the formation rate of RO2 increase by 5x, but the RO2 loss rate goes as [RO2]^2, which means that in pseudo-steady state, the RO2 concentration only changes by sqrt(5) = 2.2. Considering that the competition between unimolecular (autoxidation) and bimolecular (e.g. RO2+RO2) reactions can often differ orders of magnitude, the reported limited change with VOC increase is not sufficient evidence for the exclusion of autoxidation as possible pathway for dimer ester formation. Also, as we suggest that the bimolecular reactions of RO2 are likely involved in dimer ester formation whilst increasing VOC would lower autoxidation increased bimolecular RO2 reaction may counteract this effect on the dimer ester formation thus explaining the limited change to dimer esters as VOC conc. is increased.*

2. In a previous study (Kristensen et al., ACP, 2013, 13, 3763-3776), it was found that both the MW358 and MW368 dimers were higher under higher temperature, in both ambient measurements and chamber experiments. This is in contrast to the results of this work, making me wonder how robust the quantification of the dimer esters are. It might also be possible that the temperature affects various particle-phase chemistry that leads to the conversion between different dimer esters in different directions. The authors discussed this somewhat, but the conclusion is still unclear.

*Reply: In the study referred to by the reviewer (Kristensen et al., ACP, 2013, 13, 3763-3776) it clearly states in relation to smog chamber findings that "The pinyl-diaterpenyl ester showed somewhat higher maximum concentrations during the warm experiments (about 0.7 µg m−3 ) compared to the cold ones (about 0.4 µg m−3 ), while the pinonyl-pinyl ester showed no clear temperature dependency". Also, it is worth noting that in Kristensen et al., 2013, the warm and cold experiments were performed at 23 and 15 °C, respectively, thus considerable less extreme than in the current work. Consequently, we believe that the findings on the temperature effects on the two dimer esters in question are very much in line with our previous observations reported in 2013. With respect to ambient observations, it is stated that: "In addition, high concentrations of dimer esters in daytime samples along with a tenfold increase in the average dimer concentration in 2009 compared to 2007 suggest the presence of other factors affecting the formation of dimer esters in 2009." From the work following the 2013 paper, we now know that ozone and maybe also OH-radicals may be an important factor likely contributing to the higher dimer esters*

concentration (both pinyl-diaterpenyl and pinonyl-pinyl ester) during the warmer 2009 campaign in Blodgett Forest.

With respect to particle-phase chemistry we believe that the performed temperature ramping experiments contradict purely particle-phase formation of dimer esters because suggested dimer esters monomeric precursors (e.g. pinic acid) increase in concentration as temperature is lowered no accompanying increase in dimer esters is observed. Also, as particle-phase chemistry is most likely going to be temperature dependent (higher reaction rates at higher temperatures), we would expect increased dimer ester formation from particle chemistry as the temperature is ramped from -15 to 20 °C.

However, we recognize the reviewers concern regarding the limited discussion on particle-phase (or heterogeneous) chemistry as possibly involved in dimer ester formation as suggested in recent publications publications by Kenseth et al (PNAS, 115, 8301-8306, 10.1073/pnas.1804671115, 2018) and Zhao et al., (The Journal of Physical Chemistry A, 123, 10782-10792, 2019). Consequently we have added this to the manuscript in addition to findings regarding the possible role of OH-radicals and the above considerations on purely particle-phase chemistry as dimer ester formation pathway.

Line (613-638): "Recently, heterogeneous chemistry has been suggested as a possible pathway for the formation of dimer ester in β-pinene oxidation experiments. Here, dimer esters are suggested to form from semivolatile dicarboxylic acids (e.g. pinic acid) undergoing traditional equilibrium gas-particle partitioning with subsequent reactive uptake of the gas-phase, OH-derived monomers on collision with particle surfaces to form dimer esters (Kenseth et al., 2018). In Kenseth et al. (2018) the suggested mechanism is supported by an observed increased formation of certain dimer esters in oxidation experiments including seed particles enriched with pinic acid as well as an observed suppressed formation of some dimer esters in the presence of an OH-scavenger. In relation, Zhao et al., (2019b) recently reported dimer formation through dimerization by organic radical (i.e., peroxy, $RO_2$, and alkoxy radicals, RO) cross reactions during heterogeneous OH-initiated oxidative aging of oxygenated organic aerosol.

In the current study, heterogeneous chemistry and OH-dependency could help explain the observed response to temperature of the identified dimer esters. At the lower 0 and -15 °C reaction temperatures increased condensation of semivolatile organic acids, such as pinic acid (Fig. 5 & Fig. 7), could facilitate the formation of dimer esters from these monomeric species. In addition, reduced OH radical production at lower reaction temperatures could explain the suppressed formation of the higher O:C dimer esters from reduced formation of more oxidized organic species, such as MBTCA and DTAA (Fig. S3), and other OH-depended monomers available for reactive uptake unto the formed SOA particles or gas-phase reactions. This is somewhat supported by an observed correlations between the particle phase concentration of pinic acid and the pinonyl-pinyl ester ($R^2 = 0.92$, Fig. S6) whilst the more oxidized dimer ester pinyl-diaterpenyl ester correlate better with MBTCA ($R^2 = 0.99$, Fig. S6) rather than pinic acid ($R^2 = 0.02$). Although correlation between particle-phase monomeric organic acids and dimer esters could suggest purely particle-phase chemistry as mechanism for dimer ester formation this is not supported by the relatively small changes in the dimer ester concentration following heating or cooling of the formed SOA particles (Fig. 8). In particular, temperature-ramping experiments show that while the particle-phase concentrations of pinic acid and MBTCA are affected by both heating and cooling of the SOA particles, both pinonyl-pinyl ester and pinyl-diaterpenyl ester are unaffected with concentrations determined by the initial reaction temperatures (Fig. S6). The observed temperature effects on dimer esters formation could be ascribed to oxidation by OH-radicals of the dimer ester precursors, whether formed from autoxidation processes or $RO_2$ termination pathways, or of the dimer esters themselves thus leading to formation of the high O:C dimer esters.

**Changes made to the manuscript:**

- Line 148-149: Small changes to the original sentence.

- Line 492-496: Inserted paragraph on thermodynamic, which was originally located in the dimer ester formation section.

- Line 591-616: New paragraph on heterogeneous formation pathway

- Line 622-626: Small changes to the original sentences to include the possible role of OH-chemistry.

- Line 654-659: Small changes to the original sentences to include the possible role of OH-chemistry.

- Added two new references:

[revised manuscript text omitted]